# SAMPLE-EFFICIENT MULTI-OBJECTIVE MOLECULAR OPTIMIZATION WITH GFLOWNETS

## ABSTRACT

Many crucial scientific problems involve designing novel molecules with desired properties, which can be formulated as an expensive black-box optimization problem over the discrete chemical space. Computational methods have achieved initial success but still struggle with simultaneously optimizing multiple competing properties in a sample-efficient manner. In this work, we propose a multi-objective Bayesian optimization (MOBO) algorithm leveraging the hypernetwork-based GFlowNets (HN-GFN) as an acquisition function optimizer, with the purpose of sampling a diverse batch of candidate molecular graphs from an approximate Pareto front. Using a single preference-conditioned hypernetwork, HN-GFN learns to explore various trade-offs between objectives. Inspired by reinforcement learning, we further propose a hindsight-like off-policy strategy to share high-performing molecules among different preferences in order to speed up learning for HN-GFN. Through synthetic experiments, we illustrate that HN-GFN has adequate capacity to generalize over preferences. Extensive experiments show that our framework outperforms the best baselines by a large margin in terms of hypervolume in various real-world MOBO settings.

## 1 INTRODUCTION

Designing novel molecular structures with desired properties, also referred to as molecular optimization, is a crucial task with great application potential in scientific fields ranging from drug discovery to material design. Molecular optimization can be naturally formulated as a black-box optimization problem over the discrete chemical space, which is combinatorially large (Polishchuk et al., 2013). Recent years have witnessed the trend of leveraging computational methods, such as deep generative models (Jin et al., 2018) and combinatorial optimization algorithms (You et al., 2018; Jensen, 2019), to facilitate the optimization. However, the applicability of most prior approaches in real-world scenarios is hindered by two practical constraints: (i) realistic oracles (e.g., wet-lab experiments and high-fidelity simulations) require substantial costs to synthesize and evaluate molecules (Gao et al., 2022), and (ii) chemists commonly seek to optimize multiple properties of interest simultaneously (Jin et al., 2020b). For example, in addition to effectively inhibiting a disease-associated target, an ideal drug is desired to be easily synthesizable and non-toxic.

Bayesian optimization (BO) (Jones et al., 1998; Shahriari et al., 2015) provides a sample-efficient framework for globally optimizing expensive black-box functions. The basic idea is to construct a cheap-to-evaluate *surrogate model*, typically a Gaussian Process (GP) (Rasmussen, 2003), to approximate the true function (also known as the *oracle*) on the observed dataset. The core objective of BO is to optimize an *acquisition function* (built upon the surrogate model) in order to obtain informative candidates with high utility for the next round of evaluations. This loop is repeated until the evaluation budget is exhausted. Owing to the fact that a large batch of candidates can be evaluated in parallel in biochemical experiments, we perform batch BO (with large-batch and low-round settings (Angermueller et al., 2020)) to significantly shorten the entire cycle of optimization.

As multi-objective optimization (MOO) problems are prevalent in scientific and engineering applications, MOBO also received broad attention and achieved promising performance by effectively optimizing differentiable acquisition functions (Daulton et al., 2020). Nevertheless, it is less prominent in discrete problems, especially considering batch settings. The difficulty lies in the fact that no gradients can be leveraged to navigate the discrete space for efficient and effective optimization of

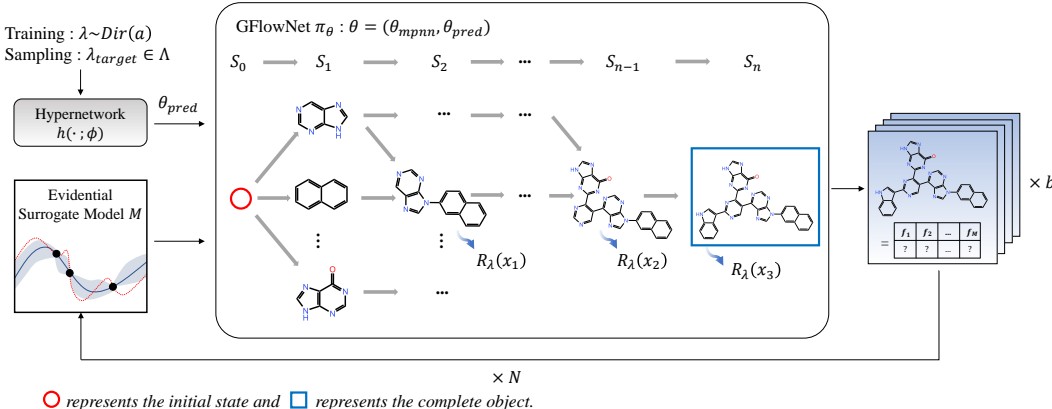

Figure 1: MOBO loop for molecular optimization using an evidential surrogate model $\mathcal{M}$ for uncertainty estimation and HN-GFN for acquisition function optimization. In each round, the policy $\pi_\theta$ is trained with reward function $R_\lambda$, where $\lambda$ is sampled from $\text{Dir}(\alpha)$ per iteration. A new batch of candidates is sampled from the approximate Pareto front according to $\lambda_{\text{target}} \in \Lambda$.

the acquisition function. Although most of the existing discrete molecular optimization methods can be adopted as the *acquisition function optimizer* to alleviate this issue, they suffer from the following limitations. 1) Most approaches do not explicitly discuss the diversity of the proposed candidates, which is a key consideration in batch settings as the surrogate model cannot exactly reproduce the oracle's full behaviors. Therefore, we not only want to cover more high modes of the surrogate model but also to obtain candidates that bring additional information about the search space. 2) Most multi-objective methods (Xie et al., 2021; Fu et al., 2022) simply rely on a scalarization function, parameterized by a predefined *preference vector* reflecting the trade-off between objectives, and turn the MOO problem into a single-objective one. Unfortunately, an ideal trade-off is unclear before optimization (even with domain knowledge), and many potential trade-offs of interest are worth exploring. In principle, it is possible to independently train multiple optimization models, each conditioned on a distinct preference vector, to cover the objective space. Practically, this trivial strategy cannot efficiently scale with the number of objectives (Navon et al., 2021).

The recently proposed GFlowNets (Bengio et al., 2021a) are a class of generative models over discrete objects (e.g., molecular graphs) that aim to learn a stochastic policy for sequentially constructing objects with a probability proportional to a reward function (e.g., the acquisition function). Hence, GFlowNets possess merit in generating diverse and high-reward objects, which makes them appealing in the batch BO context where exploration plays a significant role (Jain et al., 2022).

In this work, we present a MOBO algorithm based on GFlowNets for sample-efficient multi-objective molecular optimization. We propose a hypernetwork-based GFlowNet (HN-GFN) as the acquisition function optimizer within MOBO to sample a diverse batch of candidates from an approximate Pareto front. Instead of defining a fixed reward function as usual in past work (Bengio et al., 2021a), we train a unified GFlowNet on the distribution of reward functions (*random scalarizations* parameterized by preference vectors) and control the policy using a single preference-conditioned hypernetwork. While sampling candidates, HN-GFN explores various trade-offs between competing objectives flexibly by varying the input preference vector. Inspired by Hindsight Experience Replay (Andrychowicz et al., 2017) in RL, we further introduce a hindsight-like off-policy strategy to share high-performing molecules among different preferences and speed up learning for HN-GFN. As detailed in our reported experiments, we first evaluate HN-GFN through synthetic experiments to verify that HN-GFN is capable of generalizing over preference vectors, then apply the proposed framework to real-world scenarios. Remarkably, our framework outperforms the best baselines by 60% and 24% (relative improvement in terms of hypervolume in the settings with two and four objectives), respectively. Our key contributions are summarized below:

- We propose HN-GFN, a unified GFlowNet that can efficiently sample candidates from an approximate Pareto front using a single hypernetwork.

- We introduce a hindsight-like off-policy strategy to speed up learning in HN-GFN.
- Experiments verify that our MOBO algorithm based on HN-GFN can find high-quality Pareto front more efficiently compared to state-of-the-art baselines.

## 2 RELATED WORK

**Molecular optimization.** Recently, molecular optimization has been approached with a wide variety of computational methods, which can be mainly grouped into three categories: 1) **Latent space optimization (LSO)** methods perform the optimization over the low-dimensional continuous latent space learned by generative models such as variational autoencoders (VAEs) (Gómez-Bombarelli et al., 2018; Maus et al., 2022). These methods require the latent representations to be discriminative, but the training of the generative model is decoupled from the optimization objectives, imposing challenges for optimization (Tripp et al., 2020). Instead of navigating the latent space, combinatorial optimization methods search for the desired molecular structures directly in the explicit discrete space with 2) **evolutionary algorithms** (Jensen, 2019) and 3) **deep neural networks** to guide the searching (You et al., 2018). However, most prior methods only focus on optimizing a single property, from non-biological properties such as drug-likeness (QED) (Bickerton et al., 2012) and synthetic accessibility (SA) (Ertl & Schuffenhauer, 2009), to biological properties that measure the binding energy to a protein target (Bengio et al., 2021a). Despite the above advances, multi-objective molecular optimization has recently received wide attention (Jin et al., 2020b; Xie et al., 2021; Fu et al., 2022). For example, MARS (Xie et al., 2021) employs Markov chain Monte Carlo (MCMC) sampling to find novel molecules satisfying several properties. However, most approaches require a notoriously large number of oracle calls to evaluate molecules on-the-fly (Jin et al., 2020b; Xie et al., 2021). In contrast, we tackle this problem in a sample-efficient manner.

**GFlowNet.** GFlowNets (Bengio et al., 2021a) aim to sample composite objects proportionally to a reward function, instead of maximizing it as usual in RL (Sutton & Barto, 2018). GFlowNets are related to the MCMC methods due to the same objective, while amortizing the high cost of sampling (mixing between modes) over training a generative model (Zhang et al., 2022). GFlowNets have made impressive progress in various applications, such as active learning (Jain et al., 2022), discrete probabilistic modeling (Zhang et al., 2022), and Bayesian structure learning (Deleu et al., 2022). For a thorough discussion and mathematical treatment, we refer the readers to Bengio et al. (2021a;b)

**Bayesian Optimization for discrete spaces.** While the application of BO in continuous domains has proliferated during the last decade, effort in applying it to discrete spaces is lacking. It is much more challenging to construct surrogate models and optimize acquisition functions in discrete spaces, compared to continuous spaces. One common approach is to define GPs with discrete kernels (Moss et al., 2020) and solve the acquisition function optimization problem with evolutionary algorithms (Kandasamy et al., 2018). Moreover, AmortizedBO (Swersky et al., 2020) proposes to augment the evolutionary algorithms with RL.

**Multi-objective Bayesian Optimization.** BO has been widely used in MOO problems for efficiently optimizing multiple competing black-box functions. Most popular approaches are based on hypervolume improvement (Daulton et al., 2020), random scalarizations (Knowles, 2006; Paria et al., 2020), and entropy search (Hernández-Lobato et al., 2016). While there have been several approaches that take parallel evaluations (Bradford et al., 2018; Konakovic Lukovic et al., 2020) and diversity (Konakovic Lukovic et al., 2020) into account, they are limited to continuous domains.

## 3 BACKGROUND

### 3.1 PROBLEM FORMULATION

We address the problem of searching over a discrete chemical space $\mathcal{X}$ to find molecular graphs $x \in \mathcal{X}$ that maximize a *vector-valued objective* $f(x) = \big(f_1(x), f_2(x), \ldots, f_M(x)\big) : \mathcal{X} \to \mathbb{R}^M$, where $f_m$ is a black-box function (also known as the oracle) evaluating a certain property of molecules. Practically, realistic oracles are extremely expensive to evaluate with either high-fidelity simulations

or wet-lab experiments. We thus propose to perform optimization in as few oracle evaluations as possible, since the sample efficiency is paramount in such a scenario.

There is typically no single optimal solution to the MOO problem, as different objectives may contradict each other. Consequently, the goal is to recover the *Pareto front* – the set of *Pareto optimal* solutions which cannot be improved in any one objective without deteriorating another (Ehrgott, 2005; Miettinen, 2012). In the context of maximization, a solution $f(x)$ is said to *Pareto dominates* another solution $f(x')$ iff $f_m(x) \geq f_m(x') \, \forall m = 1, \ldots, M$ and $\exists m'$ such that $f_{m'}(x) > f_{m'}(x')$, and we denote $f(x) \succ f(x')$. A solution $f(x^*)$ is *Pareto optimal* if not Pareto dominated by any solution. The Pareto front can be written as $\mathcal{P}^* = \{f(x^*) : \{f(x) : f(x) \succ f(x^*)\} = \emptyset\}$.

The quality of a finite approximate Pareto front $\mathcal{P}$ is commonly measured by *hypervolume* (HV) (Zitzler & Thiele, 1999) – the M-dimensional Lebesgue measure $\lambda_M$ of the space dominated by $\mathcal{P}$ and bounded from below by a given reference point $\boldsymbol{r} \in \mathbb{R}^M$: $HV(\mathcal{P}, \boldsymbol{r}) = \lambda_M(\cup_{i=1}^{|\mathcal{P}|}[\boldsymbol{r}, y_i])$, where $[\boldsymbol{r}, y_i]$ denotes the hyper-rectangle bounded by $\boldsymbol{r}$ and $y_i = f(x_i)$.

## 3.2 BATCH BAYESIAN OPTIMIZATION

Bayesian optimization (BO) (Shahriari et al., 2015) provides a model-based iterative framework for sample-efficient black-box optimization. Given an observed dataset $\mathcal{D}$, BO relies on a Bayesian *surrogate model* $\mathcal{M}$ to estimate a posterior distribution over the true oracle evaluations. Equipped with the surrogate model, an *acquisition function* $a : \mathcal{X} \to \mathbb{R}$ is induced to assign the utility values to candidate objects for deciding which to next evaluate the oracle. Compared with the costly oracle, the cheap-to-evaluate acquisition function can be efficiently optimized. We consider the scenario where the oracle is given an evaluation budget of $N$ rounds with fixed batches of size $b$.

To be precise, we have access to a random initial dataset $\mathcal{D}_0 = \{(x_i^0, y_i^0)\}_{i=1}^n$, where $y_i^0 = f(x_i^0)$ is true oracle evaluation. In each round $i \in \{1, \ldots, N\}$, the acquisition function is maximized to yield a batch of candidates $\mathcal{B}_i = \{x_j^i\}_{j=1}^b$ to be evaluated in parallel on the oracle $y_j^i = f(x_j^i)$. The observed dataset $\mathcal{D}_{i-1}$ is then augmented for the next round: $\mathcal{D}_i = \mathcal{D}_{i-1} \cup \{(x_1^i, y_j^i)\}_{j=1}^b$.

## 4 METHOD

In this section, we present the proposed MOBO algorithm based on hypernetwork-based GFlowNet (HN-GFN), shown in Figure 1. Our key idea is to extend GFlowNets as the acquisition function optimizer for MOBO, with the objective to sample a diverse batch of candidates from the approximate Pareto front. To begin, we introduce GFlowNets in the context of molecule design, then describe how GFlowNet can be biased by a preference-conditioned hypernetwork to sample molecules according to various trade-offs between objectives. Next, we propose a hindsight-like off-policy strategy to speed up learning in HN-GFN. Lastly, we introduce the evidential surrogate model.

## 4.1 PRELIMINARIES

GFlowNets (Bengio et al., 2021a) seek to learn a stochastic policy $\pi$ for sequentially constructing discrete objects $x \in \mathcal{X}$ with a probability $\pi(x) \propto R(x)$, where $\mathcal{X}$ is a compositional space and $R : \mathcal{X} \to \mathbb{R}_{\geq 0}$ is a non-negative reward function. The generation process of object $x \in \mathcal{X}$ can be represented by a sequence of discrete *actions* $a \in \mathcal{A}$ that incrementally modify a partially constructed object, which is denoted as *state* $s \in \mathcal{S}$. Let generation process begin at a special initial state $s_0$ and terminate with a special action indicating that the object is complete ($s = x \in \mathcal{X}$), the construction of an object $x$ can be defined as a complete trajectory $\tau = (s_0 \to s_1 \to \cdots \to s_n = x)$.

Following fragment-based molecule design (Bengio et al., 2021a; Xie et al., 2021), we first define a vocabulary of building blocks (molecule fragments), then generate molecular graphs by sequentially attaching a fragment to an atom of the partially constructed molecules. There are multiple action sequences leading to the same state, and no fragment deleting actions, the space of possible action sequences can thus be denoted by a directed acyclic graph (DAG) $\mathcal{G} = (\mathcal{S}, \mathcal{E})$, where the edges in $\mathcal{E}$ are transitions $s \to s'$ from one state to another. To learn the aforementioned desired policy, Bengio et al. (2021a) propose to see the DAG structure as a *flow network*.

**Markovian flows.** Bengio et al. (2021b) first define a *trajectory flow* $F : \mathcal{T} \to \mathbb{R}_{\geq 0}$ on the set of all complete trajectories $\mathcal{T}$ to measure the unnormalized density. The *edge flow* and *state flow* can then be defined as $F(s \to s') = \sum_{s \to s' \in \tau} F(\tau)$ and $F(s) = \sum_{s \in \tau} F(\tau)$, respectively. The trajectory flow $F$ determines a probability measure $P(\tau) = \frac{F(\tau)}{\sum_{\tau \in \mathcal{T}} F(\tau)}$. If flow $F$ is *Markovian*, the forward transition probabilities $P_F$ can be computed as $P_F(s'|s) = \frac{F(s \to s')}{F(s)}$.

**Flow matching objective.** A flow is *consistent* if the following *flow consistency equation* is satisfied $\forall s \in \mathcal{S}$:

$$F(s) = \sum_{s' \in Pa_{\mathcal{G}}(s)} F(s' \to s) = R(s) + \sum_{s'':s \in Pa_{\mathcal{G}}(s'')} F(s \to s'') \tag{1}$$

where $Pa_{\mathcal{G}}(s)$ is a set of parents of $s$ in $\mathcal{G}$. As proved in Bengio et al. (2021a), if the flow consistency equation is satisfied with $R(s) = 0$ for non-terminal state $s$ and $F(x) = R(x) \geq 0$ for terminal state $x$, a policy $\pi$ defined by the forward transition probability $\pi(s'|s) = P_F(s'|s)$ samples object $x$ with a probability $\pi(x) \propto R(x)$. GFlowNets propose to approximate the edge flow $F(s \to s')$ using a neural network $F_\theta(s, s')$ with enough capacity, such that the flow consistency equation is respected at convergence. To achieve this, Bengio et al. (2021a) define a temporal difference-like (Sutton & Barto, 2018) learning objective, called flow-matching (FM):

$$\mathcal{L}_{\text{FM}}(s, R; \theta) = \left( \log \frac{\sum_{s' \in Pa_{\mathcal{G}}(s)} F_\theta(s', s)}{R(s) + \sum_{s'':s \in Pa_{\mathcal{G}}(s'')} F_\theta(s, s'')} \right)^2 \tag{2}$$

Bengio et al. (2021a) prove that we can use any exploratory policy $\widetilde{\pi}$ with full support to sample training trajectories and obtain the consistent flow $F_\theta(s, s')$ by minimizing the FM objective. Consequently, a policy defined by this approximate flow $\pi_\theta(s'|s) = P_{F_\theta}(s'|s) = \frac{F_\theta(s \to s')}{F_\theta(s)}$ can also sample objects $x$ with a probability $\pi_\theta(\text{x})$ proportionally to reward $R(x)$. Practically, the training trajectories are sampled from an exploratory policy which is a mixture between $P_{F_\theta}$ and a uniform distribution over allowed actions (Bengio et al., 2021a).

## 4.2 HYPERNETWORK-BASED GFLOWNETS

Our proposed HN-GFN aims at sampling a diverse batch of candidates from the approximate Pareto front with a unified model. A common approach to MOO is to decompose it into a set of scalar optimization problems with different scalarization functions and apply standard single-objective optimization methods to gradually approximate the Pareto front (Knowles, 2006; Zhang & Li, 2007). We here consider convex combinations (weighted sum) of the objectives. Let $\lambda = (\lambda_i, \cdots, \lambda_M) \in S_M$ be a preference vector defining the trade-off between the competing properties, where $S_M = \{\lambda \in \mathbb{R}^m | \sum_i \lambda_i = 1, \lambda_i \geq 0\}$ is the $M-1$ simplex. Then the scalarization function can be formulated as $s_\lambda(x) = \sum_i \lambda_i f^i(x)$.

To support parallel evaluations in BO, one can obtain candidates according to different scalarizations (Daulton et al., 2020). Practically, this approach hardly scales efficiently with the number of objectives for discrete problems. Taking GFlowNet as an example, we need to train multiple GFlowNets independently for each choice of the reward function $R_\lambda(x) = s_\lambda(x)$ to cover the objective space:

$$\theta_\lambda^* = \arg\min_\theta \mathbb{E}_{s \in \mathcal{S}} \mathcal{L}_{\text{FM}}(s, R_\lambda) \tag{3}$$

Our key motivation is to design a unified GFlowNet to sample candidates according to different reward functions, even ones not seen during training. Instead of defining the reward function with a fixed preference vector $\lambda$, we propose to train a preference-conditioned GFlowNet on a distribution of reward functions $R_\lambda$, where the preference vector $\lambda$ is sampled from a simplex $S_M$:

$$\theta^* = \arg\min_\theta \mathbb{E}_{\lambda \in S_M} \mathbb{E}_{s \in \mathcal{S}} \mathcal{L}_{\text{FM}}(s, R_\lambda) \tag{4}$$

Note that the preliminary concept of conditional GFlowNet was originally introduced in Bengio et al. (2021b). We study and instantiate this concept, aiming to facilitate MOO in the context of molecule design.

**Remarks.** Our proposed optimization scheme of training a single model to fit a family of loss functions fits into the framework of YOTO (Dosovitskiy & Djolonga, 2019). As proved in Dosovitskiy & Djolonga (2019), assuming an infinite model capacity, the proposed optimization scheme (Eq. 4) is as powerful as the original one (Eq. 3), since the solutions to both loss functions coincide. Nevertheless, the assumption of infinite capacity is extremely strict and hardly holds, so how to design the conditioning mechanism in practice becomes crucial.

### 4.2.1 HYPERNETWORK-BASED CONDITIONING MECHANISM

We propose to condition the GFlowNets on the preference vectors via hypernetworks (Ha et al., 2016). Hypernetworks are deep networks that generate the weights of a target network based on inputs. In vanilla GFlowNets, the flow predictor $F_\theta$ is parameterized with the MPNN (Gilmer et al., 2017) over the graph of molecular fragments, with two prediction heads approximating $F(s, s')$ and $F(s)$ based on the node and graph representations respectively. These two heads are parameterized with multi-layer perceptrons (MLPs).

One can view the training of HN-GFN as learning an agent to perform multiple policies that correspond to different goals (reward functions $R$) defined in the same environment (state space $\mathcal{S}$ and action space $\mathcal{A}$). Therefore, we propose to only condition the weights of prediction heads $\theta_{\text{pred}}$ with hypernetworks, while sharing the weights of MPNN $\theta_{\text{mpnn}}$, leading to more generalizable state representations. More precisely, a hypernetwork $h(\cdot; \phi)$ takes as inputs the preference vector $\lambda$ to output the weights $\theta_{\text{pred}} = h(\lambda; \phi)$ of prediction heads in the flow predictor $F_\theta$. For brevity, we write $\theta = (\theta_{\text{mpnn}}, \theta_{\text{pred}})$. Following Navon et al. (2021), we parametrize $h$ using a MLP with multiple heads, each generating weights for different layers of the target network.

### 4.2.2 AS THE ACQUISITION FUNCTION OPTIMIZER

**Training.** At each iteration, we first randomly sample a new preference vector $\lambda$ from a Dirichlet distribution $\text{Dir}(\alpha)$. Then the HN-GFN is trained in a usual manner with the reward function set as $R_\lambda(x) = a(\mu(s_\lambda(x)), \sigma(s_\lambda(x)); \mathcal{M})$, where $\mu$ and $\sigma$ are posterior mean and standard deviation estimated by $\mathcal{M}$.

**Sampling.** At each round $i$, we use the trained HN-GFN to sample a diverse batch of $b$ candidates. Let $\Lambda^i$ be the set of $l$ target preference vectors $\lambda^i_{\text{target}}$. We sample $\frac{b}{l}$ molecules per $\lambda^i_{\text{target}} \in \Lambda^i$ and evaluate them on the oracle in parallel. In practice, we simply sample $\lambda^i_{\text{target}}$ from $\text{Dir}(\alpha)$, but it is worth noting that this prior distribution can also be defined adaptively based on the trade-off of interest. As the number of objectives increases, we choose a larger $l$ to cover the objective space.

### 4.3 HINDSIGHT-LIKE OFF-POLICY STRATEGY

Resorting to the conditioning mechanism, HN-GFN can learn a family of policies to achieve various goals, i.e., one can treat sampling high-reward molecules for a particular preference vector as a separate goal. As verified empirically in Jain et al. (2022), since the FM objective is *off-policy* and *offline*, we can use offline trajectories to train the target policy for better exploration, so long as the assumption of full support holds. Our key insight is that each policy can learn from the valuable experience (high-reward molecules) of other similar policies.

To achieve this, inspired by Hindsight Experience Replay (Andrychowicz et al., 2017) in RL, we propose to share high-performing molecules among policies by re-examining them with different preference vectors. Because there are infinite possible preference vectors, here we only focus on $\Lambda^i$, which are based on to sample candidates at round $i$, and build a replay buffer for each $\lambda^i_{\text{target}} \in \Lambda^i$. After sampling some trajectories during training, we store in the replay buffers the complete object $x$ with the reward $R_{\lambda^i_{\text{target}}}(x)$.

Algorithm 2 describes the training procedure for HN-GFN with the proposed hindsight-like strategy. At each iteration, we first sample a preference vector from a mixture between $\text{Dir}(\alpha)$ and a uniform distribution over $\Lambda^i$: $(1 - \gamma)\text{Dir}(\alpha) + \gamma\text{Uniform}$. If $\Lambda$ is chosen, we construct half of the training batch with offline trajectories from the corresponding replay buffer of molecules encountered with the highest rewards. Otherwise, we incorporate offline trajectories from the current observed dataset $\mathcal{D}_i$ instead to ensure that HN-GFN samples correctly in the vicinity of the observed Pareto set.

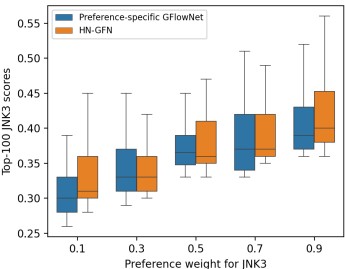 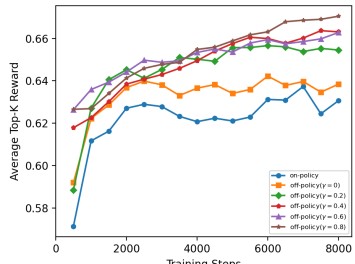

Figure 2: **Left**: The distribution of Top-100 JNK3 scores over different preference vectors. **Right**: The progression of the average Top-20 rewards over the course of training of the HN-GFN in optimizing GSK3$\beta$ and JNK3 with different strategies.

## 4.4 EVIDENTIAL SURROGATE MODEL

While GPs are well-established in continuous spaces, they scale poorly with the number of observations and do not perform well in discrete spaces (Swersky et al., 2020). There has been significant work in efficiently training non-Bayesian neural networks to estimate the uncertainty (Gal & Ghahramani, 2016). In this work, we use evidential deep learning (Amini et al., 2020) to explicitly learn the epistemic uncertainty. Compared with the widely used MC Dropout (Gal & Ghahramani, 2016) and Deep Ensembles (Lakshminarayanan et al., 2017), evidential deep learning presents the advantages of faster inference speed and superior calibrated uncertainty (Soleimany et al., 2021). As for the acquisition function, we use Upper Confidence Bound (Srinivas et al., 2010) to incorporate epistemic uncertainty. To be precise, the objectives are modeled with a single multi-task network and the acquisition function is applied to the scalarization. See Appendix C.3 for more discussion.

## 5 EXPERIMENTS

We first verify that HN-GFN has adequate capacity to generalize over preference vectors in a synthetic scenario. Next, we evaluate the effectiveness of the proposed MOBO algorithm based on HN-GFN in practical scenarios, which are more in line with real-world molecular optimization. Implementation details and additional results are provided in the Appendix.

### 5.1 SYNTHETIC SCENARIO

Here, our goal is to demonstrate that we can leverage the HN-GFN to sample molecules with preference-conditioned property distributions. The HN-GFN is used as a stand-alone optimizer outside of MOBO to directly optimize the scalarizations of oracle scores. As the oracle cannot be called as many times as necessary practically, we refer to this scenario as a synthetic scenario. To better visualize the trend of the property distribution of the sampled molecules as a function of the preference vector, we only consider two objectives: inhibition scores against glycogen synthase kinase-3 beta (GNK3$\beta$) and c-Jun N-terminal kinase-3 (JNK3) (Li et al., 2018; Jin et al., 2020b).

**Compared methods.** We compare HN-GFN against the following methods. **Preference-specific GFlowNet** is a vanilla GFlowNet trained independently for a particular preference vector. Note that the preference-specific GFlowNet is treated as "*gold standard*" rather than the baseline, as it is trained and evaluated using the same preference vector. **Concat-GFN** and **FiLM-GFN** are two variations of the conditional GFlowNet based on FiLM (Perez et al., 2018) and concatenation, respectively. **MOEA/D** (Zhang & Li, 2007) and **NSGA-III** (Deb & Jain, 2013) are two multi-objective evolutionary algorithms that also incorporate preference information. We perform evolutionary algorithms over the 32-dim latent space learned by HierVAE (Jin et al., 2020a), which gives better optimization performance than JT-VAE (Jin et al., 2018).

**Metrics.** All the above methods are evaluated over the same set of 5 evenly spaced preference vectors. For each GFlowNet-based method, we sample 1000 molecules per preference vector as

Table 1: Evaluation of different methods on the synthetic scenario

| Method | HV | Div | Cor |
|---|---|---|---|
| MOEA/D | 0.182 ± 0.045 | n/a | n/a |
| NSGA-III | 0.364 ± 0.041 | n/a | n/a |
| preference-specific GFlowNet | 0.545 ± 0.055 | 0.786 ± 0.013 | 0.653 ± 0.003 |
| Concat-GFN | 0.534 ± 0.069 | 0.786 ± 0.004 | 0.646 ± 0.008 |
| FiLM-GFN | 0.431 ± 0.045 | 0.795 ± 0.014 | 0.633 ± 0.009 |
| HN-GFN | **0.550 ± 0.074** | **0.797 ± 0.015** | **0.666 ± 0.010** |

Table 2: Evaluation of different methods on MOBO scenarios (mean ± std over 3 runs)

| | GSK3$\beta$ + JNK3 | | GSK3$\beta$ + JNK3 + QED + SA | |
|---|---|---|---|---|
| | HV | Div | HV | Div |
| HierVAE+$q$ParEGO | 0.205 ± 0.015 | n/a | 0.186 ± 0.009 | n/a |
| HierVAE+$q$EHVI | 0.341 ± 0.072 | n/a | 0.211 ± 0.006 | n/a |
| LaMOO | 0.279 ± 0.090 | n/a | 0.190 ± 0.069 | n/a |
| GP-BO | 0.368 ± 0.020 | 0.347 ± 0.059 | 0.335 ± 0.021 | 0.562 ± 0.031 |
| MARS | 0.418 ± 0.095 | 0.653 ± 0.072 | 0.273 ± 0.020 | **0.754 ± 0.027** |
| HN-GFN | 0.572 ± 0.087 | **0.810 ± 0.003** | 0.389 ± 0.012 | 0.744 ± 0.008 |
| HN-GFN w/ hindsight | **0.669 ± 0.061** | 0.793 ± 0.007 | **0.416 ± 0.023** | 0.738 ± 0.009 |

the solutions. We compare the aforementioned methods on the following metrics: **Hypervolume indicator (HV)** measures the volume of the space dominated by the Pareto front of the solutions and bounded from below by the preference point $(0,0)$. **Diversity (Div)** is the average pairwise Tanimoto distance over Morgan fingerprints. **Correlation (Cor)** is the Spearman's rank correlation coefficient between the probability of sampling molecules from an external test set under the GFlowNet and their respective rewards in the logarithmic domain (Nica et al., 2022). See more details in Appendix B.1.2. In a nutshell, HV and Div measure the quality of the solutions, while Cor measures how well the trained model is aligned with the given preference vector.

**Experimental results.** As shown in Table 1, HN-GFN outperforms the baselines and achieves competitive performance to the preference-specific GFlowNets (gold standard) on all the metrics. Compared to the GFlowNet-based methods, the evolutionary algorithms (MOEA/D and NSGA-III) fail to find high-scoring molecules, especially the MOEA/D. HN-GFN outperforms Concat-GFN and FiLM-GFN in terms of HV and Cor, implying the superiority of the well-designed hypernetwork-based conditioning mechanism. The comparable performance of HN-GFN and preference-specific GFlowNets illustrates that HN-GFN can generalize over preference vectors. Therefore, the unified HN-GFN provides a significantly efficient way to explore various trade-offs between objectives. In Figure 2 (Left), we visualize the trend of the empirical distribution of JNK3 as the preference weight increases. Intuitively, HN-GFN and preference-specific GFlowNets show consistent trends: the larger the preference weight, the higher the average score.

## 5.2 Multi-objective Bayesian optimization

Next, we evaluate the effectiveness of HN-GFN as an acquisition function optimizer within MOBO in the practical scenarios, where there is a limited evaluation budget for oracle. We consider the following objective combinations of varying size:

- GNK3$\beta$+JNK3: Jointly inhibiting Alzheimer-related targets GNK3$\beta$ and JNK3.

- GNK3$\beta$+JNK3+QED+SA: Jointly inhibiting GNK3$\beta$ and JNK3 while being drug-like and easy-to-synthesize.

We rescale the SA score such that all the above properties have a range of [0,1] and higher is better. For both combinations, we consider starting with $|\mathcal{D}_0| = 200$ random molecules and further querying the oracle $N = 8$ rounds with batch size $b = 100$.

**Baselines.** We compare HN-GFN with the following methods (from each category mentioned in section 2) as the acquisition function optimizer: **HierVAE (Jin et al., 2020a) with** $q$**ParEGO/**$q$**EHVI** (Daulton et al., 2020) and **LaMOO** (Zhao et al., 2022) are LSO methods. **GP-BO** (Tripp et al., 2021) uses Graph GA (Jensen, 2019) to optimize the acquisition function defined based on a GP with Tanimoto kernel. **MARS** (Xie et al., 2021) applies MCMC sampling to optimize the acquisition function defined based on the same surrogate function as HN-GFN. Note that the RL-based P-MOCO (Xi Lin, 2022) is also implemented but fails to optimize the properties.

**Experimental results.** Table 2 shows that HN-GFN achieves superior performance over the baselines in terms of HV and Div, especially trained with the hindsight-like off-policy strategy. Note that the Div is computed among the batch of 100 candidates per round, we omit this metric for LSO methods as they only support 160 rounds with batch size 5 due to memory constraint. Our HN-GFN w/ hindsight outperforms the best baselines MARS and GP-BO of the two objective combinations by a large margin (60.0% and 24.2% relative improvement) with respect to HV, respectively. The promising performance can be attributed to the ability of HN-GFN to sample a diverse batch of candidates from the approximate Pareto front. Another interesting observation, in the more challenging settings where four objectives are optimized, is that MARS generates diverse candidates via MCMC sampling but fails to find high-quality Pareto front, indicating that HN-GFN can find high-reward modes better than MARS. The computational costs are discussed in the Appendix B.4.

## 5.3 ABLATIONS

**Effect of the hindsight-like strategy.** In the first round of MOBO, for each $\lambda_{\text{target}} \in \Lambda$ we sample 100 molecules every 500 training steps and compute the average Top-20 reward over $\Lambda$. In Figure 2 (Right), as we vary $\gamma$ from 0 to 1, the hindsight-like strategy significantly boosts average rewards, demonstrating that sharing high-performing molecules among policies is effective for speeding up the training of HN-GFN. We choose $\gamma = 0.2$ for the desired trade-off between reward and generalization, see Appendix C.2 for a detailed explanation.

**Effect of** $\alpha$**.** Next, we study the effect of the prior distribution of preference vectors $\text{Dir}(\alpha)$. We consider the more challenging GNK3$\beta$+JNK3+QED+SA combination, where the difficulty of optimization varies widely for various properties. Table 3 shows that the distribution skewed toward harder properties results in better optimization performance.

**Effect of scalarization functions.** In addition to the weighted sum (WS), we consider the Tchebycheff (Miettinen, 2012) that is also commonly used in MOO. Table 3 shows that Tchebycheff leads to a worse Pareto front compared to WS. We conjecture that the non-smooth reward landscapes induced by Tchebycheff are harder to optimize.

Table 3: Ablation study of the $\alpha$ and scalarization functions on GNK3$\beta$+JNK3+QED+SA

|  | $\alpha$ | | | scalarization function | |
|---|---|---|---|---|---|
|  | (1,1,1,1) | (3,3,1,1) | (3,4,2,1) | WS | Tchebycheff |
| HV | 0.312 ± 0.039 | 0.385 ± 0.018 | **0.416 ± 0.023** | **0.416 ± 0.023** | 0.304 ± 0.075 |
| Div | **0.815 ± 0.015** | 0.758 ± 0.018 | 0.738 ± 0.009 | **0.738 ± 0.009** | 0.732 ± 0.014 |

## 6 CONCLUSION

We have introduced a MOBO algorithm for sample-efficient multi-objective molecular optimization. This algorithm leverages a hypernetwork-based GFlowNet (HN-GFN) to sample a diverse batch of candidates from the approximate Pareto front. In addition, we present a hindsight-like off-policy strategy to improve optimization performance. Our algorithm outperforms existing approaches on synthetic and practical scenarios. **Future work** includes extending this algorithm to other discrete optimization problems such as biological sequence design and neural architecture search.

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

# A  ALGORITHMS

Algorithm 1 describes the overall framework of the proposed MOBO algorithm, where HN-GFN is leveraged as the acquisition function optimizer. Algorithm 2 describes the training procedure for HN-GFN within MOBO.

---

**Algorithm 1** MOBO based on HN-GFN

---

**Input:** oracle $f = (f_1, \ldots, f_M)$, initial dataset $\mathcal{D}_0 = \{(x_i^0, f(x_i^0))\}_{i=1}^n$, acquisition function $a$, parameter of Dirichlet distribution $\alpha$, number of rounds $N$, batch size $b$
**Initialization:** surrogate model $\mathcal{M}$, parameters of HN-GFN $\pi_\theta$
**for** $i = 1$ **to** $N$ **do**
    Fit surrogate model $\mathcal{M}$ on dataset $\mathcal{D}_{i-1}$
    Sample the set of target preference weights $\Lambda \sim \text{Dir}(\alpha)$
    Train $\pi_\theta$ with reward function $R_\lambda(x) = a(\mu(s_\lambda(x)), \sigma(s_\lambda(x)); \mathcal{M})$     ▷ Algorithm 2
    Sample query batch $\mathcal{B}_i = \{x_j^i\}_{j=1}^b$ based on $\lambda_{target} \in \Lambda$
    Evaluate batch $\mathcal{B}_i$ with $f$ and augment the dataset $\mathcal{D}_{i+1} = \mathcal{D}_i \cup \{(x_j^i, f(x_j^i))\}_{j=1}^b$
**end for**

---

---

**Algorithm 2** Training procedure for HN-GFN with the hindsight-like off-policy strategy

---

**Input:** available dataset $\mathcal{D}_i$, reward function $R$, minibatch size $m$, set of target preference vectors $\Lambda$, proportion of hindsight-like strategy $\gamma$, replay buffers $\{\mathcal{R}_\lambda\}_{\lambda \in \Lambda}$
**while** not converged **do**
    Flag $\sim$ Bernoulli($\gamma$)
    **if** Flag = 1 **then**
        $\lambda \sim \Lambda$
        Sample $\frac{m}{2}$ trajectories from replay buffer $\mathcal{R}_\lambda$
    **else**
        $\lambda \sim \text{Dir}(\alpha)$
        Sample $\frac{m}{2}$ trajectories from the available dataset $\mathcal{D}_i$
    **end if**
    $\theta = (\theta_{\text{mpnn}}, h(\lambda; \phi))$
    Sample $\frac{m}{2}$ trajectories from policy $\widetilde{\pi}$ and store terminal states $x$ in $\mathcal{R}_\lambda$ for all $\lambda \in \Lambda$
    Compute reward $R_\lambda(x)$ on terminal states $x$ from each trajectory in the minibatch
    Update parameters $\theta_{\text{mpnn}}$ and $\phi$ with a stochastic gradient descent step w.r.t Eq. 2
**end while**

---

# B  IMPLEMENTATION DETAILS

## B.1  EXPERIMENTAL SETTINGS

### B.1.1  MOLECULE DOMAIN

Following (Bengio et al., 2021b), the molecules are generated based on a set of 105 building blocks. The same substructure containing multiple stems (atoms for linking another building block) is served as separate building blocks. We allow the GFlowNet to sample molecules with 2-8 blocks. As for the oracles, we adopt the property prediction models (random forest) released by (Xie et al., 2021) to evaluate the inhibition ability of generated molecules against GSK3$\beta$ and JNK3.

### B.1.2  METRICS

**Diversity.**   Diversity (Div) is the average pairwise Tanimoto distance over Morgan fingerprints. In the synthetic scenario, for each preference vector, we sample 1000 molecules, calculate the Div among the Top-100 molecules, and report the averages over preferences. In MOBO, the DiV is computed among the batch of 100 candidates per round, as GP-BO and MARS are not preference-conditioned. And we believe this metric possibly is more aligned with how these methods might be used in a biology or chemistry experiment.

**Correlation.** Correlation (Cor) is the Spearman's rank correlation coefficient between the probability of sampling molecules from an external test set under the GFlowNet and their respective rewards in the logarithmic domain: Cor = Spearman's $\rho_{\log(\pi(x)),\log(R(x))}$ The external test set is obtained in two steps: First, we generate a random dataset containing 300K molecules uniformly based on the number of building blocks; Next, we sample the test sets with uniform property distribution corresponding to GSK3$\beta$ and JNK3, respectively, from the 300K molecules. The final test set contains 6062 molecules.

## B.2 BASELINES

All the baselines are implemented using the publicly released source codes with adaptations for our MOBO scenarios. Our evolutionary algorithms are implemented in PyMOO (Blank & Deb, 2020), and the LSO methods are implemented in BoTorch (Balandat et al., 2020). LaMOO and GP-BO utilize EHVI as the acquisition function. For all GP-based methods, each objective is modeled by an independent GP.

## B.3 HN-GFN

We implement the proposed HN-GFN in PyTorch (Paszke et al., 2019). The values of key hyperparameters are illustrated in Table 4.

**Surrogate model:** We use the 12-layer MPNN as the base architecture of the surrogate model in our experiments. In MOBO, a single multi-task MPNN is trained with a batch size of 64 using the Adam optimizer with a dropout rate of 0.1 and a weight decay rate of 1e-6. We apply early stopping to improve generalization.

**HN-GFN:** HN-GFN contains a vanilla GFlowNet and a preference-conditioned hypernetwork. The architecture of GFlowNet is a 10-layer MPNN, and the hypernetwork is a 3-layer MLP with multiple heads, each generating weights for different layers of the target network. The HN-GFN is trained with Adam optimizer to optimize the Flow Matching objective.

## B.4 EMPIRICAL RUNNING TIME

The efficiency is compared on the same computing facilities using 1 Tesla V100 GPU. In the context of MOBO, the running time of three LSO methods (i.e., HierVAE+$q$ParEGO, HierVAE+$q$EHVI, and LaMOO) is around 3 hours, while GP-BO optimizes much faster and costs only 13 minutes. In contrast, the time complexity of deep-learning-based discrete optimization methods is much larger. MARS costs 32 hours, while our proposed HN-GFN costs 10 hours. With the hindsight-like training strategy, the running time of HN-GFN will increase roughly by 33%.

However, if we look at the problem in a bigger picture, the time costs for model training are most likely negligible in comparison to those of evaluating the molecular candidates in real-world applications. Hence, we argue that the high quality of the candidates (the performance of the MOBO algorithm) is more essential than having a lower training cost.

## C ADDITIONAL RESULTS

### C.1 SYNTHETIC SCENARIO

As illustrated in Figure 3, the distribution of Top-100 GSK3$\beta$ scores shows a consistent trend in preference-specific GFlowNet and our proposed HN-GFN, although the trend is not as significant as the JNK3 property.

### C.2 EFFECT OF THE HINDSIGHT-LIKE STRATEGY

There is a trade-off between reward and generalization. As we vary $\gamma$ from 0 to 1, the training distribution of preference vectors moves from $\text{Dir}(\alpha)$ to the set of target preference vectors $\Lambda$.

Table 4: Hyper-parameters used in the real-world MOBO experiments.

| Hyper-parameter | GSK3$\beta$ + JNK3 | GSK3$\beta$ + JNK3 + QED + SA |
|---|---|---|
| Surrogate model | | |
| Hidden size | 64 | 64 |
| Learning rate | 2.5e-4 | 1e-3 |
| $\lambda$ for evidential regression | 0.1 | 0.1 |
| Number of iterations | 10000 | 10000 |
| Early stop patience | 500 | 500 |
| Dropout | 0.1 | 0.1 |
| Weight decay | 1e-6 | 1e-6 |
| Acquisition function (UCB) | | |
| $\beta$ | 0.1 | 0.1 |
| HN-GFN | | |
| Learning rate | 5e-4 | 5e-4 |
| Reward exponent | 8 | 8 |
| Reward norm | 1.0 | 1.0 |
| Trajectories minibatch size | 8 | 8 |
| Offline minibatch size | 8 | 8 |
| hindsight $\gamma$ | 0.2 | 0.2 |
| Uniform policy coefficient | 0.05 | 0.05 |
| Hidden size for GFlowNet | 256 | 256 |
| Hidden size for hypernetwork | 100 | 100 |
| Training steps | 5000 | 5000 |
| $\alpha$ | (1,1) | (3,4,2,1) |

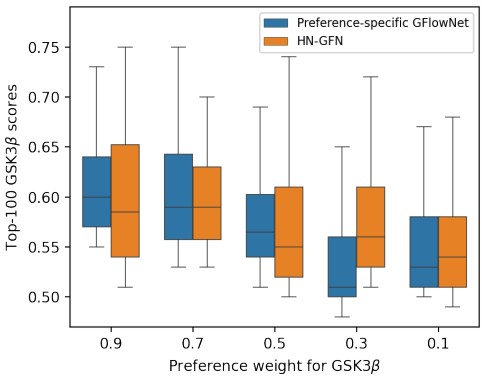

Figure 3: Comparison of the distribution of Top-100 GSK3$\beta$ scores sampled by different preference vectors using preference-specific GFlowNets and HN-GFN.

Exclusively training the HN-GFN with the finite target preference vectors can lead to poor generalization. In practice, although we only sample candidates based on $\Lambda$, we argue that it is vital to keep the generalization such that we can leverage the trained HN-GFN to explore various preference vectors adaptively. In Figure 2 (Right), we found that increasing $\gamma$ leads to slight (not significant) improvement in average reward compared to $\gamma = 0.2$. Hence, we believe 0.2 is the desired trade-off.

## C.3 EFFECT OF SURROGATE MODELS

We conduct ablation experiments to study the effectiveness of different surrogate models. We consider the following three surrogate models: evidential regression (Amini et al., 2020), Deep Ensembles (Lakshminarayanan et al., 2017), and GP based on Tanimoto kernel (Tripp et al., 2021). As

shown in Table 5, we can observe that evidential regression leads to better optimization performance than Deep Ensembles. While the HV of evidential regression and GP is comparable, evidential regression can propose more diverse candidates. Furthermore, we argue that GP is less flexible over discrete spaces than evidential regression and Deep Ensembles, as different kernels need to be designed according to the data structures.

Table 5: Evaluation of different surrogate models on MOBO scenarios

|  | GSK3$\beta$ + JNK3 | | GSK3$\beta$ + JNK3 + QED + SA | |
|---|---|---|---|---|
|  | HV | Div | HV | Div |
| HN-GFN (Evidential) | **0.669 ± 0.061** | 0.793 ± 0.007 | 0.416 ± 0.023 | 0.738 ± 0.009 |
| HN-GFN (Ensemble) | 0.583 ± 0.103 | **0.797 ± 0.004** | 0.355 ± 0.048 | **0.761 ± 0.012** |
| HN-GFN (GP) | 0.662 ± 0.054 | 0.739 ± 0.008 | **0.421 ± 0.037** | 0.683 ± 0.018 |

## C.4 SAMPLED MOLECULES IN MOBO EXPERIMENTS

We give some examples of sampled molecules from the Pareto front by HN-GFN in the GSK3$\beta$ + JNK3 + QED + SA optimization setting (Figure 4). The numbers below each molecule refer to GSK3$\beta$, JNK3, QED, and SA scores respectively.

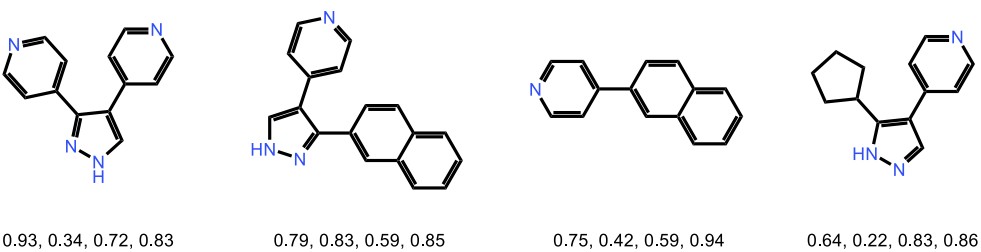

| 0.93, 0.34, 0.72, 0.83 | 0.79, 0.83, 0.59, 0.85 | 0.75, 0.42, 0.59, 0.94 | 0.64, 0.22, 0.83, 0.86 |

Figure 4: Sampled molecules from the approximate Pareto front by HN-GFN.

