# OpenReview forum: "Sample-efficient multi-objective molecular optimization with GFlowNets"
_ICLR.cc/2023/Conference — Submitted to ICLR 2023_

### Official Review · Reviewer_4QvB · 2022-10-20

**Confidence:** 4
**Correctness:** 3
**Technical Novelty And Significance:** 2
**Empirical Novelty And Significance:** 2
**Recommendation:** 3

**Clarity, Quality, Novelty And Reproducibility:**

## Clarity & Quality
Most parts of the paper are clearly written. Some details are missing, though, for example $\gamma$.

## Novelty
The idea is actually not entirely, but has been long mentioned as "Pareto GFlowNet" in [GFlowNet Foundations](https://arxiv.org/pdf/2111.09266.pdf).

## Reproducibility
I cannot check the reproducibility as it seems the implementation is not provided.

**Strength And Weaknesses:**

## Strengths
1. The methodology is well-formulated, with careful design for amortizing multi-objective problems and surrogate modeling.
2. The off-policy property of GFlowNet is examined, including a mix of different training trajectory distribution, and the use of hindsight replay idea.

## Weaknesses
1. I would expect some illustration / analysis on at least toy tasks to show that the proposed method indeed learn the multi-objective behavior. E.g., when the conditioning preference is changed, the distribution of GFlowNet will change accordingly.
2. The diversity should be measured for every fixed preference. If allowed different preference, there will be of course diverse samples generated by the GFlowNet.
3. Is there any particular reason to model the preference conditioning with a hypernetwork?  Hypernetwork is not very stable in training. What's more, usually it is enough to directly augment the input with preference for the conditioning, which is also much easier. I would expect the authors to justify this choice with empirical evidence.
4. Is there any particular reason to use evidental regression? According to my experience, sometimes it is not as reliable as more "traditional" methods like deep ensemble. Also, it seems HierVAE-based methods and GP-BO still uses GP, while the proposed method uses evidental regression. Is this an unfair comparison? I would expect at least some results about using HN-GFN with GP to justify this choice.

### Minors
- The experiments are limited to molecule generation. The multi-objective problems are much more than small graph generation. Other applications, such as protein, sequences, are also of great importance.
- Is the set of target preference vectors $\Lambda$ kept fixed? If not, how to update it?
- In Fig.2 (left), why HN-GFN could be better than the "gold-standard" preference-specific GFlowNet? Do these two use the same GFlowNet architecture?
- Fig.2 (right) shows the performance with $\gamma=0, 0.2$, but it would be great to see a spectrum from 0 to 1. This is important for the choice of hyper parameter $\gamma$, which seems not mentioned in the paper. Please correct me if I miss anything!


**Summary Of The Paper:**

This work investigate the possibility of using GFlowNets to tackle multi-objective sampling problems. Specifically, hypernetwork-based GFlowNet is proposed for solving multi-objective Bayesian optimization problems. Some insights from reinforcement learning is also involved. The proposed algorithm is evaluated in molecule generation tasks.

**Summary Of The Review:**

The method of this paper is clean and straightforward, while this work still suffers from non-extensive experiments and insufficient ablation to justify some usage of the components. I understand that it is a tight schedule for the authors to rebuttal, but I would of course consider raising the score if some of the main concerns are addressed.

---

> ### Author Response · Authors · 2022-11-17
> **Response to Reviewer 4QvB (1/2)**
>
> We deeply appreciate the reviewer for the insightful and constructive comments.
>
> > I would expect some illustration / analysis on at least toy tasks to show that the proposed method indeed learns the multi-objective behavior. E.g., when the conditioning preference is changed, the distribution of GFlowNet will change accordingly.
>
> Indeed, the main purpose of the synthetic experiment (Section 5.1) is to show that the proposed preference-conditioned GFlowNet can learn the multi-objective behavior and compare the effectiveness of different conditioning mechanisms. We have revised our unclear description (the first two sentences in Section 5.1) to emphasize this purpose. Fig.2 (left) shows that the property distribution of the molecules sampled by HN-GFN will change as the conditioning preference changes and the trend is consistent with the preference-specific GFlowNets, indicating HN-GFN indeed learns the multi-objective behavior. We hope this clarification clears this concern.
>
> > The diversity should be measured for every fixed preference. If allowed different preferences, there will be of course diverse samples generated by the GFlowNet.
>
> Thanks for pointing this out, we agree with your comment. In Section 5.1, we have updated the metrics in Table 1. While the new metrics are slightly worse than the original metrics indeed, all GFlowNet-based methods still maintain comparable and high diversity. In Section 5.2, our purpose is to verify that we can sample a diverse batch of molecules (regardless of the preference conditions). Hence, we still choose to report the original metrics for a fair comparison, as GP-BO and MARS are not preference-conditioned, and possibly more aligned with how these methods might be used in a biology or chemistry experiment. To avoid ambiguity, we have detailed metrics in Appendix B.1.2.
>
> > Is there any particular reason to model the preference conditioning with a hypernetwork? Hypernetwork is not very stable in training. What's more, usually it is enough to directly augment the input with preference for the conditioning, which is also much easier. I would expect the authors to justify this choice with empirical evidence.
>
> In our experiments, hypernetwork is quite stable. We attribute this to our design where only the weights of prediction heads are conditioned with hypernetwork while the weights of the encoder are shared among all preference vectors (as detailed in the second paragraph in Section 4.2.1). Based on the advice, we have added another conditional GFlowNet (Concat-GFN), which directly augments the input with preference by concatenation, as a baseline in Section 5.1. As shown in Table 1, HN-GFN outperforms Concat-GFN and FiLM-GFN.
>
> |   |  HV | Div | Cor |
> | :-----| ----: | ----: | ----: |
> | FiLM-GFN | 0.431 ± 0.045 | 0.795 ± 0.014 | 0.633 ± 0.009 |
> | Concat-GFN | 0.534 ± 0.069 | 0.786 ± 0.004 | 0.646 ± 0.008 |
> | preference-specific GFlowNet | 0.545 ± 0.055 | 0.786 ± 0.013 | 0.653 ± 0.003 |
> | HN-GFN | **0.550 ± 0.074** | **0.797 ± 0.015** | **0.666 ± 0.010** |
>
> > Is there any particular reason to use evidential regression? According to my experience, sometimes it is not as reliable as more "traditional" methods like deep ensemble. Also, it seems HierVAE-based methods and GP-BO still uses GP, while the proposed method uses evidential regression. Is this an unfair comparison? I would expect at least some results about using HN-GFN with GP to justify this choice.
>
> Since GP is less flexible over discrete spaces (different kernels, e.g., graph kernel and string kernel, need to be designed according to the data structures), increasing discrete optimization methods use neural networks to estimate uncertainty recently [1]. There are two reasons to use evidential regression: 1) It enables faster calculation of rewards. 2) Soleimany et al. [2] have demonstrated that evidential uncertainties enable superior calibrated predictions for molecular property prediction than deep ensemble and MC dropout. We have conducted ablation experiments to study the effectiveness of different surrogate models. We can observe that Evidential leads to better optimization performance than Ensemble. While the HV of Evidential and GP is comparable, Evidential can propose more diverse candidates. Additionally, Evidential is more flexible and general, and can be directly used for other discrete optimization problems. We have discussed these results in Appendix C.3.
>
> |  | GSK3β + JNK3 |  |  GSK3β + JNK3 + QED + SA |  |
> | :-----| ----: | ----: | ----: | ----: |
> |   |  HV | Div | HV | Div |
> |  HN-GFN (Evidential) |  **0.669 ± 0.061**  | 0.793 ± 0.007 | 0.416 ± 0.023 | 0.738 ± 0.009 |
> | HN-GFN (Ensemble) |  0.583 ± 0.103 | **0.797 ± 0.004** | 0.355 ± 0.048 | **0.761 ± 0.012** |
> | HN-GFN (GP) | 0.662 ± 0.054 | 0.739 ± 0.008 | **0.421 ± 0.037** | 0.683 ± 0.018 |
>
> [1] Amortized Bayesian Optimization over Discrete Spaces
>
> [2] Evidential Deep Learning for Guided Molecular Property Prediction and Discovery

---

> > ### Author Response · Authors · 2022-11-17
> > **Response to Reviewer 4QvB (2/2) Minor**
> >
> > > The experiments are limited to molecule generation. The multi-objective problems are much more than small graph generation. Other applications, such as protein, sequences, are also of great importance.
> >
> > Thanks for your advice, we agree that multi-objective protein design is also a very important scientific problem. As we mentioned in the conclusion, we leave it for future work.
> >
> > > Is the set of target preference vectors Λ kept fixed? If not, how to update it?
> >
> > It is a great question, we are sorry for our unclear description and we have revised the Section 4.2.2. The set of target preference vectors Λ is updated every BO round. At each round, we simply sample target preference vectors from $\operatorname{Dir}(\alpha)$. Put differently, the target preference vectors and training preference vectors follow the same distribution.
> >
> > > In Fig.2 (left), why HN-GFN could be better than the "gold-standard" preference-specific GFlowNet? Do these two use the same GFlowNet architecture?
> >
> > Our main purpose is to show the consistent trends between HN-GFN and preference-specific GFlowNets, see Response for the first comment for a more detailed explanation. There are two explanations for Fig.2: 1) Since the top-100 scores are computed by sampling, we do not see a significant difference between them and we think they are comparable. 2) our HN-GFN may share knowledge across preference vectors using a single hypernetwork, while preference-specific GFlowNet is trained with a fixed preference vector. Yes, they use the same GFlowNet architecture for a fair comparison.
> >
> > > Fig.2 (right) shows the performance with  $\gamma$=0,0.2, but it would be great to see a spectrum from 0 to 1. This is important for the choice of hyperparameter  $\gamma$, which seems not mentioned in the paper. Please correct me if I miss anything!
> >
> > We thank the reviewer for pointing this out. We have updated Fig.2 (right) to show a spectrum from 0 to 1 (interval 0.2). We think that there is a trade-off between reward and generalization. As we vary $\gamma$ from 0 to 1, the training distribution of preference vectors moves from $\operatorname{Dir}(\alpha)$ to the set of target preference vectors Λ. Exclusively training the HN-GFN with the finite target preference vectors can lead to poor generalization. In practice, although we only sample candidates based on Λ, we argue that it is vital to keep the generalization such that we can leverage the trained HN-GFN to explore various preference vectors adaptively. We found that increasing  $\gamma$ leads to slight (not significant) improvement in average reward compared to  $\gamma$=0.2. Hence, we believe 0.2 is a reasonable trade-off.
> >
> > We hope our response can alleviate your concerns. Please let us know if you have any additional questions.

---

> > > ### Comment · Reviewer_4QvB · 2022-12-11
> > > **Thank you for the response**
> > >
> > > Thank you for the detailed response. I appreciate the effort for providing explanations to the questions. However, after reading the answers, I feel my original concern about the weak performance of the proposed method is validated. Therefore, I will keep the original score.

---

> > > > ### Author Response · Authors · 2022-12-12
> > > > **Response to Reviewer 4QvB**
> > > >
> > > > Thank you for engaging in the discussion! Would you mind elaborating on the answers that validate your concern, such that we can further improve our work?
> > > >
> > > > We want to emphasize that our main purpose is to tackle practical challenges and constraints of molecular optimization in real-world scenarios, and our key contribution is to study and instantiate the concept of conditional GFlowNet and propose a GFlowNet-based framework to facilitate multi-objective optimization. In addition, we also delicately propose a hindsight-like off-policy strategy to speed up training. More importantly, our framework is general and agnostic to the conditioning mechanisms and surrogate models. Put differently, our framework can be coupled with various conditioning mechanisms and surrogate models based on specific application scenarios and data. Therefore, we argue that the comparison of different components in the framework should be a demonstration of the generality of our framework, rather than downplaying our contribution.
> > > >
> > > > We believe these are significant contributions in the context of developing the nascent framework, GFlowNet, and real-world molecular optimization applications. We hope you can re-examine our work based on the above explanations. Please let us know if you have any additional questions.

---

> ### Author Response · Authors · 2022-11-24
> **Looking forward to your feedback**
>
> Dear Reviewer 4QvB,
>
> We hope it does not disturb you. Thanks again for your insightful and constructive comments. We hope our responses have addressed your concerns. Please let us know if you have any further questions. We would be happy to answer them.
>
> Best regards,
>
> Authors

---

> ### Author Response · Authors · 2022-12-06
> **Looking forward to your feedback**
>
> Dear Reviewer 4QvB,
>
> We hope it does not disturb you. Thanks again for your insightful and constructive comments. As we are approaching the end of the rebuttal phase, we would be grateful to receive further feedback from you. We hope our responses have addressed your concerns. Please let us know if you have any further questions. We would be happy to answer them.
>
> Best regards,
>
> Authors

---

> ### Author Response · Authors · 2022-12-11
> **Looking forward to your feedback**
>
> Dear Reviewer 4QvB,
>
> We hope it does not disturb you. Thanks again for your insightful and constructive comments. As there is only last 1 day left until the end of the rebuttal phase, we would be grateful to receive further feedback from you. We hope our responses have addressed your concerns. We believe that we have improved the quality of the manuscript based on the comments raised by you and other reviewers. We are eager to engage in any further discussions if needed.
>
> Best regards,
>
> Authors

---

### Official Review · Reviewer_ZgdY · 2022-10-24

**Confidence:** 3
**Correctness:** 3
**Technical Novelty And Significance:** 3
**Empirical Novelty And Significance:** 4
**Recommendation:** 8

**Clarity, Quality, Novelty And Reproducibility:**

Clarify: The paper is well written in general, but some of the algorithmic details can be better explained, as I discussed above.

Quality: The algorithmic design and the experiments are of high quality.

Novelty: The problem of multi-objective BO for molecular optimization is intuitive and hence not novel, but the use of the hypernetwork to condition on different preference vectors in GFlowNets is novel as far as I know.

Reproducibility: Some experimental details are discussed, but the code is not uploaded.

**Strength And Weaknesses:**

Strengths:
- The proposed method is intuitive and modifies GFlowNets in a reasonable way to facilitate multi-objective optimization.
- The experiments are nicely done, and the experimental results look promising.
- The paper is in general well written, and the proposed method is well motivated.

Weaknesses:
- I feel that some of the algorithmic details are not clearly explained, particularly the connection between Algorithm 1 (for training HN-GFN) and BO. For example, does the dataset $\mathcal{D}$ correspond to the currently available observations from all previous iterations of BO? Does the reward function $R$ here correspond to the acquisition function calculated in BO? How is the set of target preference vectors built? More importantly, do you need to run Algorithm 1 after every iteration (or every batch) of BO? If yes, then the computational costs may become an issue and hence should discussed.
- I think Section 4.3 needs to be revised to make it clearer, in the current form, it's not easy to understand.
- Top paragraph of page 2: it's still unclear to me why limitation 1) makes "existing discrete molecular optimization methods" not applicable as "acquisition function optimizer". Can't you simply use those acquisition functions which directly take diversity into account? For example, if you use the GP-BUCB acquisition function from paper [1] below, to select an input in a batch, you can simply invoke an existing discrete molecular optimization method to maximize the acquisition function (whose GP posterior standard deviation is updated every time a new input is selected), which will naturally lead to a diverse set of inputs in a batch.
[1] Parallelizing Exploration-Exploitation Tradeoffs in Gaussian Process Bandit Optimization, JMLR 2014
- Top paragraph of page 9: the number of rounds $N=8$ is in fact unusually small in BO, and the batch size $b=100$ is also unusually large for BO as well. Are these choices the common practice in molecular optimization using GFlowNet?
- (minor) In the Related Work section, the previous works on multi-objective BO should also be discussed.


**Summary Of The Paper:**

This paper proposes a method for multi-objective Bayesian optimization for molecular optimization. The proposed method uses GFlowNets to optimize the acquisition function in BO, and uses a hypernetwork-based method to incorporate the preference vector into GFlowNets such that a diverse set of points can be sampled from the Pareto front.

**Summary Of The Review:**

The paper solves an important problem for molecular optimization using GFlowNets, and I don't have major concerns about the paper. The concerns I listed under "Weaknesses" are mostly regarding the writing of the paper.

---

> ### Author Response · Authors · 2022-11-17
> **Response to Reviewer ZgdY**
>
> We deeply appreciate the reviewer for the insightful and constructive comments!
>
> > I feel that some of the algorithmic details are not clearly explained, particularly the connection between Algorithm 1 (for training HN-GFN) and BO. For example, does the dataset D correspond to the currently available observations from all previous iterations of BO? Does the reward function R here correspond to the acquisition function calculated in BO? How is the set of target preference vectors built? More importantly, do you need to run Algorithm 1 after every iteration (or every batch) of BO? If yes, then the computational costs may become an issue and hence should discussed.
>
> Thanks for pointing this out, we are sorry for the unclear description. Fortunately, your understanding about the algorithm is correct. We have summarized the algorithms for MOBO and HN-GFN in Appendix A to clearly connect them. At each round, we simply sample target preference vectors from $\operatorname{Dir}(\alpha)$. Put differently, the target preference vectors and training preference vectors follow the same distribution. We have revised the Section 4.2.2. In principle, we retrain HN-GFN after every round. And we also tried just initializing the parameters of hypernetwork. We found that the performance of both approaches is similar. We have added a discussion about the time complexity in the context of MOBO in Appendix B.4. As we discussed, in many real-world settings, we argue that the high quality of the candidates (the performance of the MOBO algorithm) is more essential than having a lower training cost.
>
> > I think Section 4.3 needs to be revised to make it clearer, in the current form, it's not easy to understand.
>
> Thanks for pointing this out. We have carefully revised this section to make it clearer.
>
> > Top paragraph of page 2: it's still unclear to me why limitation 1) makes "existing discrete molecular optimization methods" not applicable as "acquisition function optimizer". Can't you simply use those acquisition functions which directly take diversity into account? For example, if you use the GP-BUCB acquisition function from paper [1] below, to select an input in a batch, you can simply invoke an existing discrete molecular optimization method to maximize the acquisition function (whose GP posterior standard deviation is updated every time a new input is selected), which will naturally lead to a diverse set of inputs in a batch.
> > [1] Parallelizing Exploration-Exploitation Tradeoffs in Gaussian Process Bandit Optimization, JMLR 2014
>
> It’s a great question, and we thank you for providing this related paper. Since no gradients can be leveraged for discrete optimization problems, most existing discrete molecular optimization methods (including the vanilla GFlowNet) need to train a specific model for a given objective function. Therefore, they are not suitable for optimizing dynamic objectives (the posterior standard deviation is updated every time) due to inefficiency. In contrast, we can sample a diverse batch of candidates with a single trained HN-GFN.
>
> > Top paragraph of page 9: the number of rounds $N=8$ is in fact unusually small in BO, and the batch size $b=100$ is also unusually large for BO as well. Are these choices the common practice in molecular optimization using GFlowNet?
>
> Yes. While these choices are unusual for common problems in BO, they are common but challenging in molecular optimization and biological sequence design, where large-batch and low-round setting is desired, as candidates can be evaluated in parallel in biochemical experiments. For example, DyNA PPO [1] considers 10 rounds with a batch size of 100. The original paper of GFlowNet [2] even considers a batch size of 200.
>
> [1] Model-based reinforcement learning for biological sequence design, ICLR 2020
>
> [2] Flow Network based Generative Models for Non-Iterative Diverse Candidate Generation, NeurIPS 2021
>
> > In the Related Work section, the previous works on multi-objective BO should also be discussed.
>
> Thanks for your constructive advice. We have updated our paper to add a discussion about MOBO in Section 2
>
> We hope our response can alleviate your concerns. Please let us know if you have any additional questions,

---

### Official Review · Reviewer_44K1 · 2022-10-24

**Confidence:** 4
**Correctness:** 3
**Technical Novelty And Significance:** 2
**Empirical Novelty And Significance:** 2
**Recommendation:** 6

**Clarity, Quality, Novelty And Reproducibility:**

**Clarity**

For the most part the discussion in the paper is clear. However, there are several places where prior work is incorrectly cited / missed out completely. In addition to the preference-conditional GFlowNet mentioned in the previous section, on line 4 on page 4, the authors cite Daulton et al. 2020 for the basic terminology of MOO instead of classic work in MOO [1,2]. On the second line in the paragraph before equation 3 the authors cite Daulton et al 2020 again for  "To support parallel evaluations in BO, one can obtain candidates according to different scalarizations", however, Daulton et al. 2020 does not consider scalarization at all. In fact it is other work [3] which establishes such approaches. Aside from this, there are aspects of the method which are also not clear. For example, the authors mention they use UCB as the acquisition function, but do not discuss how the UCB is used in this multi-objective setting - is the UCB applied to each objective individually or to the scalarization?

**Quality and Novelty**

While some of the underlying ideas are not particularly novel - preference conditioning was introduced in [4] and GFlowNets in the context of BO was studied in [5] - the hypernetwork-based conditioning and hindsight experience replay are novel in the context of GFlowNets. However, as discussed in the previous section the empirical evidence is not substantial enough.

**Reproducibility**

The authors do not provide code with the submission. The appendix does contain some relevant hyperparameters but some implementation details are not discussed at all. For example, no details are discussed about the training of the surrogate model - for instance whether the surrogate for each property is trained independently or a single multi-task model is trained, as well as other training details. The authors also do not mention other important hyperparameters like the UCB parameter (controlling exploration and exploitation).


[1] Multicriteria optimization, Ehrgott, 2005

[2] Nonlinear Multiobjective Optimization, Miettinen, 2012

[3] A Flexible Framework for Multi-Objective Bayesian Optimization using Random Scalarizations

[4] GFlowNet Foundations

[5] Biological Sequence Design with GFlowNets

**Strength And Weaknesses:**

**Strengths**

- The paper tackles an important and challenging problem of multi-objective optimization in the context of molecule generation. As shown in previous work, using GFlowNets for optimizing the acquisition function results in diverse candidates and sample-efficient optimization
- The hypernetwork-based approach is an interesting way to implement conditioning, in contrast to FiLM based approaches.

**Weaknesses**
- The paper overall is not very clearly written (I discuss this in more detail in the next section)
- The paper uses the preference-conditional GFlowNet formulation originally proposed in [1], but does not cite the paper where the preference-conditional GFlowNet is introduced. There are also other other inconsistent citations which I discuss in the next section,
- Multi-Objective REINFORCE [2] is very closely related to the proposed HN-GFN approach, differing only in the learning objective, but is not discussed / included as a baseline. Additionally the authors also do not include recent approaches such as LaMOO [3] in the baselines.
- Aside from the baselines, the experiments seem somewhat limited. While the method enjoys superior performance in the task studied in the paper, it is not clear how well it generalizes to different settings and even different rewards for instance. The authors also provide only limited ablations to investigate the method. For instance it is not clear how the set of preference vectors is selected and what is the effect of the distribution of preference vectors used in training.
- Minor: The authors use evidential regression for the surrogate model claiming "evidential deep learning presents the
advantages of faster inference speed and superior calibrated uncertainty" however, recent work [4] has established that such approaches can be arbitrarily miscalibrated.


[1]  - GFlowNet Foundations

[2] - Pareto Set Learning for Neural Multi-Objective Combinatorial Optimization

[3] - Multi-objective Optimization by Learning Space Partitions

[4] - Pitfalls of Epistemic Uncertainty Quantification through Loss Minimisation

**Summary Of The Paper:**

Practical molecule generation involve optimization of multiple objectives simultaneously. These objectives are often expensive to evaluate making sample-efficiency key. The paper proposes a multi-objective Bayesian optimization method leveraging GFlowNets for optimizing the acquisition function. GFlowNets learn stochastic policies to generate discrete objects proportionally to their rewards, resulting in diverse candidates. The authors consider a preference-based decomposition of the MOO problem. The paper proposes a hypernetwork-based parameterization for conditioning on preferences. The authors also propose a hindsight-experience replay based strategy for leveraging offline data during learning. The authors use an evidential regressor as the surrogate model within the multi-objective Bayesian optimization context. The authors then present results on a molecule generation task with 4 objectives.

**Summary Of The Review:**

In summary, while the paper presents an interesting GFlowNet-based approach to tackle multi-objective optimization, there are several major shortcomings in the paper in terms of the empirical analysis, baselines and framing of contributions. In the current state I lean towards rejection but encourage the authors to incorporate the feedback to improve the paper during the discussion.

---

> ### Author Response · Authors · 2022-11-17
> **Response to Reviewer 44K1 (1/2)**
>
> We deeply appreciate the reviewer for the insightful and constructive comments!
>
> **Clarity**
> > The paper overall is not very clearly written.
>
> We thank the reviewer for a careful and detailed review report. We have updated our paper to prudently fix the inconsistent citations accordingly:
>
> > On line 4 on page 4, the authors cite Daulton et al. 2020 for the basic terminology of MOO instead of classic work in MOO.
>
> Thanks for pointing this out. We cite Ehrgott 2005 and Miettinen 2012 in the revised paper.
>
> > On the second line in the paragraph before equation 3 the authors cite Daulton et al 2020 again for "To support parallel evaluations in BO, one can obtain candidates according to different scalarizations", however, Daulton et al. 2020 does not consider scalarization at all. In fact it is other work [1] which establishes such approaches.
>
> The reason we cite Daulton et al. 2020 is that they extend scalarization-based ParEGO [2] and TS-TCH [1] to support parallel evaluations and regard them as baselines. However, parallel evaluations are not considered in [1]. Nevertheless, we agree that [1] is an important literature about MOBO, and we have introduced it in the added paragraph ‘Multi-objective Bayesian optimization’ in Section 2.
>
> > There are aspects of the method which are also not clear. For example, the authors mention they use UCB as the acquisition function, but do not discuss how the UCB is used in this multi-objective setting - is the UCB applied to each objective individually or to the scalarization?
>
> We appreciate this very helpful critique. The UCB is applied to the scalarization, and we have added more details in Section 4.4. In addition, we have also thoroughly revised our writing according to the comments raised by you and other reviewers, please see the overall response for a summary of the revised writing.
>
> **Weaknesses**
>
> > The paper uses the preference-conditional GFlowNet formulation originally proposed in [3], but does not cite the paper where the preference-conditional GFlowNet is introduced.
>
> We thank the reviewer for pointing this out. We have added a discussion about [3] after equation 4 (page 5).
>
> > Multi-Objective REINFORCE is very closely related to the proposed HN-GFN approach, differing only in the learning objective, but is not discussed / included as a baseline. Additionally, the authors also do not include recent approaches such as LaMOO in the baselines.
>
> We did not find the source code of LaMOO when we implemented the baselines. The source code of LaMOO was not released until 19 Sep 2022 (the latest version on arXiv), which is 10 days before the deadline of ICLR 2023. Now, we have implemented LaMOO and added the results in Table 2. As for Multi-objective REINFORCE, we adapted the code to our settings but found that this approach fails to optimize the molecular properties, so the metrics are not reported. This phenomenon was also reported in [4] where reinforcement learning (PPO) fails in the multi-round molecule discovery task. We have added this description in the paragraph ‘Baselines’ in Section 5.2.
>
> |  | GSK3β + JNK3 | GSK3β + JNK3 + QED + SA |
> | :-----| ----: | ----: |
> |  | HV | HV |
> | LaMOO   | 0.279 ± 0.090  | 0.190 ± 0.069 |
> | HN-GFN  | **0.669 ± 0.061** | **0.416 ± 0.023** |
>
> [1] A Flexible Framework for Multi-Objective Bayesian Optimization using Random Scalarizations.
>
> [2] ParEGO: A Hybrid Algorithm With On-Line Landscape Approximation for Expensive Multiobjective Optimization Problems
>
> [3] GFlowNet Foundations
>
> [4] Flow Network based Generative Models for Non-Iterative Diverse Candidate Generation

---

> > ### Author Response · Authors · 2022-11-17
> > **Response to Reviewer 44K1 (2/2)**
> >
> > > The experiments seem somewhat limited. While the method enjoys superior performance in the task studied in the paper, it is not clear how well it generalizes to different settings and even different rewards for instance. The authors also provide only limited ablations to investigate the method. For instance, it is not clear how the set of preference vectors is selected and what is the effect of the distribution of preference vectors used in training.
> >
> > We appreciate this useful critique. We have done more ablation studies to provide further insights on our proposed extension of GFlowNet. These additional ablation experiments focus on 1) **Scalarization functions** (Section 5.3). In addition to the weighted sum (WS), we consider the Tchebycheff approach. Tchebycheff leads to a worse Pareto front compared to WS. We conjecture that the non-smooth reward landscapes induced by Tchebycheff are harder to optimize. 2) **Prior distribution of preference vectors used in training**. We have revised Section 4.2.2 for a clearer description about how the set of preference vectors is selected. Specifically, at each round, we simply sample target preference vectors from $\operatorname{Dir}(\alpha)$. Put differently, the target preference vectors and training preference vectors follow the same distribution. We have studied the effect of the distribution in the added Section 5.3. We found that the distribution skewed toward harder properties results in better optimization performance. 3) **Ratio of hindsight trajectories**. We have discussed it in Section 5.3.
> >
> > **scalarization functions**
> >
> > |  | WS | Tchebycheff |
> > | :-----| ----: | ----: |
> > |  HV |  **0.416 ± 0.023** |  0.304 ± 0.075 |
> > | Div   |  **0.738 ± 0.009** | 0.732 ± 0.014 |
> >
> > $\alpha$
> > |  | (1,1,1,1) | (3,3,1,1) |  (3,4,2,1) |
> > | :-----| ----: | ----: | ----: |
> > |  HV |  0.312 ± 0.039 | 0.385 ± 0.018 | **0.416 ± 0.023** |
> > | Div   | **0.815 ± 0.015** |  0.758 ± 0.018 | 0.738 ± 0.009 |
> >
> > > The authors use evidential regression for the surrogate model claiming "evidential deep learning presents the advantages of faster inference speed and superior calibrated uncertainty" however, recent work has established that such approaches can be arbitrarily miscalibrated.
> >
> > We thank the reviewer for providing this related paper, we will reconsider and analyze the uncertainty in light of their results. There are two reasons to use evidential regression: 1) It enables faster calculation of rewards. 2) Soleimany et al. [5] have demonstrated that evidential uncertainties enable superior calibrated predictions for molecular property prediction than deep ensemble and MC dropout. We have conducted ablation experiments to study the effectiveness of different surrogate models (discussed in Appendix C.3). We can observe that Evidential leads to better results than Ensemble. While the performance of Evidential and GP is comparable, Evidential is more flexible and general, and can be directly used for other discrete optimization problems. Specifically, GP is less flexible over discrete spaces (different kernels, e.g., graph kernel and string kernel, need to be designed according to the data structures). Recently, increasing discrete optimization methods use a neural network to estimate uncertainty [6].
> >
> > |  | GSK3β + JNK3 |  |  GSK3β + JNK3 + QED + SA |  |
> > | :-----| ----: | ----: | ----: | ----: |
> > |   |  HV | Div | HV | Div |
> > |  HN-GFN (Evidential) |  **0.669 ± 0.061**  | 0.793 ± 0.007 | 0.416 ± 0.023 | 0.738 ± 0.009 |
> > | HN-GFN (Ensemble) |  0.583 ± 0.103 | **0.797 ± 0.004** | 0.355 ± 0.048 | **0.761 ± 0.012** |
> > | HN-GFN (GP) | 0.662 ± 0.054 | 0.739 ± 0.008 | **0.421 ± 0.037** | 0.683 ± 0.018 |
> >
> > **Reproducibility**
> >
> > > The authors do not provide code with the submission. The appendix does contain some relevant hyperparameters but some implementation details are not discussed at all. For example, no details are discussed about the training of the surrogate model - for instance whether the surrogate for each property is trained independently or a single multi-task model is trained, as well as other training details. The authors also do not mention other important hyperparameters like the UCB parameter (controlling exploration and exploitation).
> >
> > Thanks for pointing this out. A single multi-task model is trained to estimate all properties. $\beta = 0.1$ in UCB. We have shown more details in Appendix B.3.
> >
> > We hope our response can alleviate your concerns. Please let us know if you have any additional questions.
> >
> > [5] Evidential Deep Learning for Guided Molecular Property Prediction and Discovery.
> >
> > [6] Amortized Bayesian Optimization over Discrete Spaces.

---

> > > ### Comment · Reviewer_44K1 · 2022-11-21
> > > **Response to rebuttal.**
> > >
> > > Thank you for taking into account my feedback, and the detailed response.
> > >
> > > **Clarity**: Most of my major concerns have been addressed, however, the paper still lacks some detail in exposition as pointed out by other reviewers as well.
> > >
> > > **Baselines**: Thank you for adding the baselines. I am still somewhat surprised that MOReinforce (PMOCO) does fails to optimize the rewards. Have you tried using your own implementation with just a modified learning objective? (i.e. GFlowNet flow matching -> REINFORCE policy gradient)
> > >
> > > **Ablations**: Thank you for adding the ablation. I do not think the ablation on $\alpha$ is designed properly, First, as the authors stated here the train and test distributions are kept the same - but the point of the ablation was to find the effect of varying the preference distribution and changing the target as well reveals little about that. Second, the values of $\alpha$ chosen are not justified and do not reveal much either (no coefficients <0 for example).
> > >
> > > **Surrogate Model**: I appreciate the ablation provided by the authors in support of the claim. It is certainly very useful. I would still encourage authors to make the claims about evidential regression weaker and cite Bengs et al.
> > >
> > > Overall, I believe the rebuttal answers a lot of the questions I had and improved the paper quite a bit. I have increased my score to reflect that.

---

> > > > ### Author Response · Authors · 2022-12-06
> > > > **Response to Reviewer 44K1**
> > > >
> > > > Thank you for engaging in the discussion and providing further insightful comments!
> > > >
> > > > > Most of my major concerns have been addressed, however, the paper still lacks some detail in exposition as pointed out by other reviewers as well.
> > > >
> > > > Thanks for pointing this out. We will further revise our writing to make the exposition clearer.
> > > >
> > > > > Thank you for adding the baselines. I am still somewhat surprised that MOReinforce (PMOCO) does fails to optimize the rewards. Have you tried using your own implementation with just a modified learning objective? (i.e. GFlowNet flow matching -> REINFORCE policy gradient)
> > > >
> > > > Yes, we have modified our implementation to optimize the objective of MOReinforce. We have further tuned the hyperparameters but it still fails. Besides, we found that other RL algorithms such as PPO also failed, even on the simplified problem where the model is trained using a fixed preference (mean). We conjecture that the failure is caused by the unstable training of RL and the difficulty of molecular optimization. We will continue to study this issue.
> > > >
> > > > > Thank you for adding the ablation. I do not think the ablation on $\alpha$ is designed properly, First, as the authors stated here the train and test distributions are kept the same - but the point of the ablation was to find the effect of varying the preference distribution and changing the target as well reveals little about that. Second, the values of  $\alpha$ chosen are not justified and do not reveal much either (no coefficients <0 for example).
> > > >
> > > > Thanks for pointing this out. In our early experiments, we tried different values of $\alpha$ (0.2, 0.5, 1, 1.5) and found that their performance is similar. Hence, we think our model has adequate capacity to generalize over preference vectors. Then, we found that the distribution of the target preference vectors is more important as the difficulty of optimization varies widely for various properties. We will revise the description to make our purpose clear. Moreover, the parameters $\alpha$ of the Dirichlet distribution must be greater than 0. We are sorry for our unclear definition, and we will add a discussion about the Dirichlet distribution.
> > > >
> > > >
> > > > >  I appreciate the ablation provided by the authors in support of the claim. It is certainly very useful. I would still encourage authors to make the claims about evidential regression weaker and cite Bengs et al.
> > > >
> > > > Thanks for your constructive advice. We will cite Bengs et al. and make the claims about evidential regression weaker in Section 4.4.
> > > >
> > > > We hope our response can alleviate your concerns. Please let us know if you have any additional questions.

---

> > > > > ### Comment · Reviewer_44K1 · 2022-12-10
> > > > > **Response**
> > > > >
> > > > > > Yes, we have modified our implementation to optimize the objective of MOReinforce. We have further tuned the hyperparameters but it still fails. Besides, we found that other RL algorithms such as PPO also failed, even on the simplified problem where the model is trained using a fixed preference (mean). We conjecture that the failure is caused by the unstable training of RL and the difficulty of molecular optimization. We will continue to study this issue.
> > > > >
> > > > > I find it a bit surprising that even with a fixed preference vector RL approaches don't work, since it essentially becomes a single objective optimization task on which previous work [1] that this paper builds upon has shown PPO to work. I would encourage the authors to review their implementation.
> > > > >
> > > > > > Thanks for pointing this out. In our early experiments, we tried different values of (0.2, 0.5, 1, 1.5) and found that their performance is similar. Hence, we think our model has adequate capacity to generalize over preference vectors. Then, we found that the distribution of the target preference vectors is more important as the difficulty of optimization varies widely for various properties. We will revise the description to make our purpose clear. Moreover, the parameters of the Dirichlet distribution must be greater than 0. We are sorry for our unclear definition, and we will add a discussion about the Dirichlet distribution.
> > > > >
> > > > > My apologies - I mistakenly mentioned $\alpha < 0$ instead of $\alpha < 1$ in my response. I think it is a good idea to have  all the values tried as the ones in the paper don't convey anything useful.
> > > > >
> > > > > Thanks for the clarifications. I will keep my score.
> > > > >
> > > > >
> > > > > [1] - https://arxiv.org/abs/2106.04399

---

> > > > > > ### Author Response · Authors · 2022-12-11
> > > > > > **Response to Reviewer 44K1**
> > > > > >
> > > > > > Thanks for your detailed and helpful feedback!
> > > > > >
> > > > > > > I find it a bit surprising that even with a fixed preference vector RL approaches don't work, since it essentially becomes a single objective optimization task on which previous work [1] that this paper builds upon has shown PPO to work. I would encourage the authors to review their implementation.
> > > > > >
> > > > > > In [1], PPO only works on single-round experiments (Sec 4.2) but does not work on multi-round experiments (Sec 4.3), which is similar to our setting. They claimed that 'PPO training was unstable and diverged consistently so the numbers are not reported' (in paragraph ‘Small Molecules' in Sec 4.3). We will further analyze the reasons for this phenomenon.
> > > > > >
> > > > > > [1] - https://arxiv.org/abs/2106.04399
> > > > > >
> > > > > > > My apologies - I mistakenly mentioned  $\alpha < 0$ instead of $\alpha < 1$  in my response. I think it is a good idea to have all the values tried as the ones in the paper don't convey anything useful.
> > > > > >
> > > > > > Thanks for pointing this out, we will add results and explanation about the $\alpha$ we tried in the early experiments (skewed distribution with $\alpha < 1$ and $\alpha >1$).
> > > > > >
> > > > > > We hope our response can alleviate your concerns. Please let us know if you have any additional questions.

---

### Official Review · Reviewer_NQij · 2022-11-04

**Confidence:** 3
**Correctness:** 2
**Technical Novelty And Significance:** 2
**Empirical Novelty And Significance:** 2
**Recommendation:** 5

**Clarity, Quality, Novelty And Reproducibility:**

The novelty is limited since the paper uses a combination of previous approaches.

Reproducibility is also concerning since the paper reports only three runs.

**Strength And Weaknesses:**

Strengths:
+ The paper presents an important scientific application.
+ The paper presents some promising experimental results, though I have some reservations about the robustness of the results
+ The paper is easy to follow
+ The paper addresses the multi-objective problem, which is relatively less studied in the context of molecules, but it is worth noting that it has been recently extensively studied in the general Bayesian optimization problem

Weaknesses

+ The proposed technique is a direct combination of existing techniques with no new substantial addition. Therefore, the technical contribution and novelty are weak
+ The paper uses the following statement “We assume that the oracle can be called as many times as necessary.” In Expensive settings, this is not usually true.
+ The paper does not provide any time complexity analysis of the training. This is problematic because existing approaches are actually very fast, while gflownet is certainly much more expensive, so a time comparison and a discussion about complexity and tradeoffs are important.
+ The paper discusses the sampling of the scalars extensively, but later in experiments, it is mentioned that 5 evenly-spaced preference vectors are used. It is not clear how this works exactly.
+ State of the art
    - State-of-the-art methods in molecular optimization are not stated or compared to.  The following is considered SOTA work and covers a wide range of relevant methods and benchmarks that should be discussed for fairness. [1]
    - Most multi-objective BO papers are not mentioned nor compared to. In batch optimization the most efficient and high-performing methods are [2,3,4]. It is also misleading to state that no previous paper discussed diversity while [2] is a diversity-oriented method, and several single objective batch BO papers discussed diversity.
    - The paper uses a scalarization technique where scalars are sampled from a distribution. This technique was previously proposed and used [5].
    - The Preference-based problem has been studied beyond the discussion that was mentioned in the paper. It is concerning to completely ignore principled existing work and claim it as a novelty. The following are some of the approaches, to name a few [6,7,8]

+ The experimental setup is weak and surprising:
- the paper states in the beginning that the evaluation was on several synthetic experiments and real-world experiments, but there are only two experiments.
    - The paper uses three runs only to report the mean and standard error. BO papers report AT LEAST 10 runs usually, and most recent papers report 50 to 100 runs. I don’t think 3 runs can provide any statistical significance or deliver any conclusions about performance. In fact, even expensive deep learning models are usually tested with a higher number of runs.
    - The diversity is not reported for some of the algorithms. If diversity is a metric applied to the Pareto front, why can’t it be applied to some of the approaches?
    - The paper reports results for batch size 100 only. Batch BO papers usually evaluate several batch sizes.
    - There is a total absence of many relevant baselines from molecular optimization, Bayesian optimization, and preference-based optimization. The paper is mainly experimental since the technical novelty is weak. Therefore, it needs to present a thorough experimental evaluation.

[1] Maus, Natalie, et al. "Local Latent Space Bayesian Optimization over Structured Inputs." arXiv preprint arXiv:2201.11872 (2022).

[2] Konakovic Lukovic, Mina, Yunsheng Tian, and Wojciech Matusik. "Diversity-guided multi-objective bayesian optimization with batch evaluations." Advances in Neural Information Processing Systems 33 (2020): 17708-17720.

[3] Eric Bradford, Artur M Schweidtmann, and Alexei Lapkin. Efficient multiobjective optimization employing gaussian processes, spectral sampling and a genetic algorithm. Journal of global optimization, 71(2):407–438, 2018.

[4] Syrine Belakaria and Aryan Deshwal. Uncertainty-aware search framework for multi-objective bayesian optimization. In AAAI Conference on Artificial Intelligence (AAAI), 2020.

[5] Paria, Biswajit, Kirthevasan Kandasamy, and Barnabás Póczos. "A flexible framework for multi-objective bayesian optimization using random scalarizations." Uncertainty in Artificial Intelligence. PMLR, 2020.

[6] Abdolshah M, Shilton A, Rana S, Gupta S, Venkatesh S. Multi-objective Bayesian optimization with preferences over objectives. Advances in neural information processing systems. 2019;32.

[7] Taylor, Kendall, et al. "Bayesian preference learning for interactive multi-objective optimization." Proceedings of the Genetic and Evolutionary Computation Conference. 2021.

[8] Lin, Zhiyuan Jerry, et al. "Preference Exploration for Efficient Bayesian Optimization with Multiple Outcomes." International Conference on Artificial Intelligence and Statistics. PMLR, 2022.



**Summary Of The Paper:**

The paper proposes a multi-objective Bayesian optimization approach for the molecules design problem. The proposed approach uses the hypernetwork-based GFlowNets as an acquisition function optimizer and uses a scalarization approach to combine the multiple objectives.


**Summary Of The Review:**

The paper addresses an important problem however, the novelty and experimental setup are limited.

---

> ### Author Response · Authors · 2022-11-17
> **Response to Reviewer NQij (1/3)**
>
> We appreciate your critical comments on our paper. However, we believe some of the concerns are caused by potential misunderstandings, and we hope that our response can address your concerns to some extent.
>
> > The proposed technique is a direct combination of existing techniques with no new substantial addition. Therefore, the technical contribution and novelty are weak.
>
> First of all, we want to emphasize that we primarily focus on optimization over **discrete** spaces. As we discussed in Section 2 (the second sentence in ‘molecular optimization’), while molecular optimization can be cast as a continuous optimization problem via generative models, the unsupervised training of the generative model is decoupled from the optimization objectives, leading to a latent space that is not discriminative. Hence, the performance of latent space optimization (LSO) methods is limited (shown in Section 5.2).
>
> While BO and MOBO are well-designed for problems in continuous spaces, most of them are less effective in discrete spaces. Therefore, it is not trivial to adapt some basic existing techniques in BO for discrete optimization problems (We detail this in response to comments about the state of the art).
>
> More importantly, the main purpose of our work is to tackle the practical challenges and constraints of molecular optimization in real-world scenarios, and our key technical contribution is to extend GFlowNets to facilitate multi-objective optimization:
> - We propose a preference-conditioned GFlowNet (HN-GFN) to sample diverse candidates from the approximate Pareto front, and we use the hypernetwork-based conditioning mechanism.
> - We delicately propose a hindsight-like off-policy strategy to speed up training in HN-GFN for multi-objective optimization.
>
> We believe these are significant contributions in the context of developing the nascent framework, GFlowNet.
>
> > The paper uses the following statement “We assume that the oracle can be called as many times as necessary.” In Expensive settings, this is not usually true.
>
> We agree, and this is why we refer to this scenario as a synthetic scenario in our original manuscript. Our purpose is to verify that our model can generalize over the space of preference vectors to learn the multi-objective behavior and compare the effectiveness of different conditioning mechanisms. Based on the observation that a single conditional HN-GFN achieves competitive performance to the "gold-standard" preference-specific GFlowNets, we then leverage it as the acquisition function optimizer in MOBO (Section 5.2). We have revised our unclear description (the first two sentences in Section 5.1) to remove ambiguity and emphasize our purpose.
>
> > The paper does not provide any time complexity analysis of the training. This is problematic because existing approaches are actually very fast, while GFlowNet is certainly much more expensive, so a time comparison and a discussion about complexity and tradeoffs are important.
>
> Thanks for pointing this out. We have added a discussion about the time complexity in the context of MOBO in Appendix B.4. Our HN-GFN is slower than LSO methods (if the training time of the generative model, which provides the latent space, is not included) and evolutionary algorithms, but our method is more efficient than another deep-learning-based method MARS. However, if we look at the problem in a bigger picture, the time costs for model training are most likely negligible in comparison to those of evaluating the molecular candidates in real-world applications. Hence, we argue that the high quality of the candidates (the performance of the MOBO algorithm) is more essential than having a lower training cost.
>
> > The paper discusses the sampling of the scalars extensively, but later in experiments, it is mentioned that 5 evenly-spaced preference vectors are used. It is not clear how this works exactly.
>
> For training of conditional GFlowNets (i.e., HN-GFN and two baselines Concat-GFN and FiLM-GFN), we sample a preference vector from a prior Dirichlet distribution at each iteration. Note that these trained models can generalize to handle unseen preference vectors. In contrast, preference-specific GFlowNets and two evolutionary algorithms can only perform optimization w.r.t. a pre-defined set of preference vectors. Therefore, we evaluate all the methods over the same set of preference vectors, which is used by preference-specific GFlowNets and evolutionary algorithms, for a fair comparison. We have moved the sentence ‘All the above methods are evaluated over the same set of 5 evenly spaced preference vectors.’ to the paragraph ‘metrics’ for a clearer description of the evaluation.

---

> > ### Author Response · Authors · 2022-11-17
> > **Response to Reviewer NQij (2/3) State of the art**
> >
> > > State-of-the-art methods in molecular optimization are not stated or compared to. LOL-BO is considered SOTA work and covers a wide range of relevant methods and benchmarks that should be discussed for fairness.
> >
> > Thanks for providing this related paper. As we emphasized in the first response, we focus on optimization over discrete spaces. However, LOL-BO and the relevant methods covered are all single-objective LSO methods. In addition, the covered single-objective benchmarks are not suitable for our MOO method. Nevertheless, we have cited it in Section 2 where LSO is introduced. To enhance our comparison with SOTA multi-objective molecular optimization methods, we have added the LaMOO [1] as a baseline and reported the metrics in Table 2. Our method outperforms all LSO baselines by a large margin.
> >
> > |  | GSK3β + JNK3 | GSK3β + JNK3 + QED + SA |
> > | :-----| ----: | ----: |
> > |  | HV | HV |
> > | HierVAE+qParEGO | 0.205 ± 0.015 | 0.186 ± 0.009 |
> > | HierVAE+qEHVI | 0.341 ± 0.072 | 0.211 ± 0.006 |
> > | LaMOO   | 0.279 ± 0.090  | 0.190 ± 0.069 |
> > | HN-GFN  | **0.669 ± 0.061** | **0.416 ± 0.023** |
> >
> > [1] Zhao, Yiyang, et al. "Multi-objective Optimization by Learning Space Partitions." International Conference on Learning Representations. 2022.
> >
> > > Most multi-objective BO papers are not mentioned nor compared to. In batch optimization the most efficient and high-performing methods are DGEMO, TSEMO, and USeMO. It is also misleading to state that no previous paper discussed diversity while DGEMO is a diversity-oriented method, and several single objective batch BO papers discussed diversity.
> >
> > We appreciate the reviewer’s helpful reminder, and have updated our paper to discuss about MOBO in Related Work (Section 2). It is not our intention to leave a wrong impression by omitting DGEMO and similar methods. Rather our focus was on the optimization over the discrete space and we did not want to get readers confused about this distinction by citing too many papers which we could not conveniently benchmark in this study. Please note that while the aforementioned methods (by the reviewer) take batch optimization and diversity into account, they are limited to continuous spaces as is given.
> >
> > > The paper uses a scalarization technique where scalars are sampled from a distribution. This technique was previously proposed and used in TS-TCH.
> > >
> > > The Preference-based problem has been studied beyond the discussion that was mentioned in the paper. It is concerning to completely ignore principled existing work and claim it as a novelty. The following are some of the approaches, to name a few [1,2,3]
> > >
> > >  [1] Abdolshah M, Shilton A, Rana S, Gupta S, Venkatesh S. Multi-objective Bayesian optimization with preferences over objectives. Advances in neural information processing systems. 2019;32.
> > >
> > > [2] Taylor, Kendall, et al. "Bayesian preference learning for interactive multi-objective optimization." Proceedings of the Genetic and Evolutionary Computation Conference. 2021.
> > >
> > > [3] Lin, Zhiyuan Jerry, et al. "Preference Exploration for Efficient Bayesian Optimization with Multiple Outcomes." International Conference on Artificial Intelligence and Statistics. PMLR, 2022.
> >
> > As pointed out in Section 4.2 (the second sentence) in our original manuscript, the scalarization technique (parameterized by preference vectors) is widely used in MOO, even earlier than TS-TCH and the mentioned preference-based methods given by the reviewer. Hence, we might have not exhaustively cited all relevant papers, but we certainly did not ignore all works either. Furthermore, we did not claim this scalarization technique as a novelty (by us) anywhere in this work. Rather our focus and contributions are strictly framed in the context of further developing GFlowNet such that it could be used in a more variety of applications and scenarios. Although these techniques are not novel, we believe that extending GFlowNet to MOO with these techniques is, collectively, novel and promising.

---

> > > ### Author Response · Authors · 2022-11-17
> > > **Response to Reviewer NQij (3/3) Experiments**
> > >
> > > > The paper uses three runs only to report the mean and standard error. BO papers report AT LEAST 10 runs usually, and most recent papers report 50 to 100 runs. I don’t think 3 runs can provide any statistical significance or deliver any conclusions about performance. In fact, even expensive deep learning models are usually tested with a higher number of runs.
> > >
> > > We have carefully checked multiple optimization papers to discuss this. Our findings are: 1) Most BO methods over low-dimensional continuous space report more than 10 runs (20 to 100). 2) Most **single-round** DL-based discrete optimization methods report 5 or 10 runs [1]. 3) Most **multi-round** DL-based discrete optimization methods report 3 runs [2]. Our method falls into the third category. Furthermore, previous GFlowNet-related papers [3,4] also consistently adopt the convention of three runs.
> > >
> > > [1] Xie, Yutong, et al. "MARS: Markov Molecular Sampling for Multi-objective Drug Discovery." International Conference on Learning Representations. 2021.
> > >
> > > [2] Swersky, Kevin, et al. "Amortized bayesian optimization over discrete spaces." Conference on Uncertainty in Artificial Intelligence. PMLR, 2020.
> > >
> > > [3] Bengio, Emmanuel, et al. "Flow network based generative models for non-iterative diverse candidate generation." Advances in Neural Information Processing Systems 34, (2021)
> > >
> > > [4] Malkin, Nikolay, et al. "Trajectory Balance: Improved Credit Assignment in GFlowNets." arXiv preprint arXiv:2201.13259 (2022).
> > >
> > > > The diversity is not reported for some of the algorithms. If diversity is a metric applied to the Pareto front, why can’t it be applied to some of the approaches?
> > >
> > > The diversity is applied to the sampled molecules per batch with the objective to verify that our method can sample a diverse batch of candidates. In Section 5.1, it is not reported for evolutionary algorithms as there are very few non-dominated solutions. In Section 5.2, as we discussed in the second sentence in paragraph ‘Experimental results’, for a fair comparison, we omit it as LSO methods only support 160 rounds with batch size 5 for memory constraint.
> > >
> > > > The paper reports results for batch size 100 only. Batch BO papers usually evaluate several batch sizes.
> > >
> > > We think that the reason why batch BO papers usually evaluate several batch sizes is that these algorithms are unstable when the batch size increases and can hardly scale to a larger batch size (e.g., more than 100). However, after training, our HN-GFN proposes candidates by sampling (the sampling cost is linear in the batch size). As there is no standard benchmark for sample-efficient multi-objective molecular optimization, we believe that 8 rounds and total 1000 evaluation budgets are reasonable and affordable in practice, and our experimental results convincingly show that this setting is enough to find valuable molecules.
> > >
> > > > There is a total absence of many relevant baselines from molecular optimization, Bayesian optimization, and preference-based optimization. The paper is mainly experimental since the technical novelty is weak. Therefore, it needs to present a thorough experimental evaluation.
> > >
> > > For the technical novelty, please see all responses above. For a more thorough experimental evaluation, please see the overall response for a summary of the added experiments.
> > >
> > > We hope our response can alleviate your concerns. Please let us know if you have any additional questions,

---

> ### Author Response · Authors · 2022-11-24
> **Looking forward to your feedback**
>
> Dear Reviewer NQij,
>
> We hope it does not disturb you. Thanks again for your critical comments on our paper. We believe some of the concerns are caused by potential misunderstandings. Above all, it is not trivial to adapt some basic existing techniques in BO for **discrete optimization problems** that are our focus. We hope our responses have addressed your concerns. Please let us know if you have any further questions. We would be happy to answer them.
>
> Best regards,
>
> Authors

---

> ### Author Response · Authors · 2022-12-06
> **Looking forward to your feedback**
>
> Dear Reviewer NQij,
>
> We hope it does not disturb you. Thanks again for your critical comments on our paper. As we are approaching the end of the rebuttal phase, we would be grateful to receive further feedback from you. We believe some of the concerns are caused by potential misunderstandings. Above all, it is not trivial to adapt some basic existing techniques in BO for discrete optimization problems that are our focus. We hope our responses have addressed your concerns. Please let us know if you have any further questions. We would be happy to answer them.
>
> Best regards,
>
> Authors

---

> > ### Comment · Reviewer_NQij · 2022-12-09
> > **Reviewer response**
> >
> > Thank you for the detailed rebuttal. I believe a few points that I raised were addressed, especially with respect to the claimed contribution and differences in terms of baselines for multi-objective optimization. However, I don't think that adapting previous techniques as baselines is as challenging as it is claimed to be. I will increase my score, but I still believe that the paper lacks technical novelty and thorough experimental evaluation and comparisons.

---

> > > ### Author Response · Authors · 2022-12-11
> > > **Response to Reviewer NQij**
> > >
> > > Thank you for engaging in the discussion and providing further insightful comments! As we focus on directly performing the optimization over discrete spaces, we only include several representative multi-objective latent space optimization (LSO) methods as baselines. We will implement more LSO baselines based on your comments. Please let us know if you have any additional questions.

---

### Author Response · Authors · 2022-11-17
**Overall response**

Dear all reviewers,

We deeply appreciate the reviewers for the insightful and constructive comments. We provide responses for the individual comments in the replies to the individual reviewer. We hope that our response can address the concerns raised by the reviewers to some extent.

Based on the feedback from all reviews, we have mainly made the following changes in the revised manuscript (highlighted in red):

**Writing**
1. We have added a discussion about Multi-objective Bayesian optimization in Related Work (Section 2).
2. We have revised Section 4.2.2 for a clear description about how the set of target preference vectors is selected.
3. We have revised Section 4.3 to make the hindsight-like off-policy strategy clearer.
4. We have revised the first three sentences in Section 5.1 for a clearer purpose of the synthetic experiments.
5. We have summarized the algorithms for HN-GFN and MOBO in Appendix A.
6. We have added a discussion about the computational costs in Appendix B.4.
7. We have included the Appendix at the end of the main pdf after the references, instead of leaving it as a separate file in the supplementary material.

**Experiments**
1. Add Concat-GFN (Section 5.1) and LaMOO (Section 5.2) as baselines.
2. Study the effects of the hindsight-like strategy with different ratios (Section 5.3).
3. Study the effects of different prior distributions $\operatorname{Dir}(\alpha)$ (Section 5.3).
4. Study the effects of different scalarization functions (Section 5.3)
5. Study the effects of different surrogate functions (Appendix C.3)

---

### Author Response · Authors · 2022-12-06
**Gentle reminder for feedback of our response**

Dear Reviewers,

We hope it does not disturb you. Thanks again for your insightful and constructive comments. As we are approaching the end of the rebuttal phase, we would be grateful to receive further feedback from you. We hope our responses have addressed your concerns. Please let us know if you have any further questions. We would be happy to answer them.

Best regards,

Authors

---

### Decision · Program_Chairs · 2023-01-20

**Decision:**

Reject

**Justification For Why Not Higher Score:**

N/A

**Justification For Why Not Lower Score:**

N/A

**Metareview: Summary, Strengths And Weaknesses:**

This paper considers the problem of molecule design to optimize multiple objectives with an emphasis on evaluating a batch of molecules in each round. The solution employs the hypernetwork-based GFlowNets as acquisition function optimizer and scalarization based acquisition function. This is a very important problem.

There is a lot of work on Bayesian optimization over 1) combinatorial spaces; and 2) multi-objective optimization. So a natural approach would be to combine the best methods from #1 and #2 in a synergistic manner. Some candidates are combining state-of-the-art latent space BO methods (e.g., LOL-BO, LADDER) from #1 and "Diversity-guided multi-objective bayesian optimization with batch evaluations." Advances in Neural Information Processing Systems 33 (2020): 17708-17720 from #2. Similarly, GP based surrogate models using fingerprint kernel with USeMO from #2. The paper has some candidate baselines, but they may not be the best ones available in this spirit and/or employed appropriately (e.g., qEHVI works very well in practice)

In this problem setting, we want to see the sample-efficiency of the approach. The typical evaluation procedure to show performance curves with hypervolume or diversity on y-axis and number of evaluated molecules on x-axis. In the past, researchers have shown that RL based approaches are not sample efficient for optimization problems with expensive evaluations. These performance curves are missing in the paper.

The novelty of the technical approach is somewhat limited as it heavily relies on leveraging prior methods. The related work on BO over combinatorial spaces and multi-objective BO does not reflect the current state of knowledge -- lot of missing references (some of them are pointed out by the reviewers). Please see the reference list from the recent NeurIPS-2022 tutorial on Bayesian Optimization. Additionally, there should be some qualitative discussion of why the proposed method should perform better than a Bayesian optimization solution, which is missing.

The paper is promising, but it falls short of acceptance for the above reasons. Hence, I recommend rejecting the paper. I strongly encourage the authors' to improve the paper based on the review feedback for resubmission.